# Perceived deservingness shapes attitudes toward environmental migrants in rural Bangladesh
Lukas Rudolph [1] ✉, Linus Hormuth [2], Jan Freihardt [3] & Vally Koubi[3]

As climate change intensifies, internal migration due to extreme climate events is becoming increasingly common in the Global South. Yet, little is known about how rural host communities respond to incoming environmental migrants. Here, we study attitudes toward environmentally displaced people in northern Bangladesh, focusing on perceived deservingness, empathy through shared experience, and exploratory proxy indicators of prior migrant exposure/contact. Using a pre-registered face-to-face survey of 265 rural residents, including a forced-choice conjoint experiment, we assess how migrant characteristics (reason for migration, occupation, religion, distance to origin) affect host community attitudes. We find that migrants displaced by riverbank erosion are more likely to be accepted than economic migrants (by 21%-points, $p < 0.01$) and face less discrimination based on other characteristics, indicating that deservingness strongly shapes attitudes. Regarding shared experience of erosion, which we propose as a proxy for empathy, models estimated a positive coefficient (13%-points, $p = 0.122$), hence not supporting, but indicative of a positive association between experiential proximity and greater acceptance of environmental migrants. We find no credible evidence for heterogeneity in migrant acceptance using coarse proxy measures of prior migrant exposure/contact. These results suggest that, even in resource-constrained regions, moral judgments play a central role, and that experiential proximity may be associated with more inclusive attitudes, informing policies for societal resilience under environmental stress.

The accelerating pace of climate change is fundamentally reshaping patterns of human mobility, particularly in the Global South. The Intergovernmental Panel on Climate Change (IPCC) warns that climate-related risks are intensifying faster than previously projected, with extreme weather events such as floods, storms, and droughts increasingly displacing vulnerable populations[1]. Although much of this displacement occurs within national borders, it is poised to grow in scope and complexity over the coming decades[2]. As internal climate migration intensifies, understanding how host communities perceive and respond to incoming migrants becomes critical for anticipating socio-political dynamics in regions facing concurrent environmental and demographic pressures.

Scholarly attention to climate-induced migration has expanded strongly in recent years. Research has examined the environmental drivers of migration[3–6], the interplay between climate vulnerability and mobility[7,8], and the governance of displacement and resettlement[9,10]. Yet, in contrast to this growing literature on the causes and trajectories of environmental migration, there remains a notable gap in our understanding of how host communities, particularly in the Global South, interpret and react to climate-displaced populations. This is a critical oversight as host attitudes fundamentally shape the political, economic, and social environments that determine migrants' access to resources, security, and long-term integration[11–14].

While studies in high-income countries have begun to explore public opinion toward environmental (climate) migrants[15–20], the dynamics of host attitudes in the Global South, where the majority of climate-induced migration is expected to occur, remain underexplored. Moreover, much of the existing empirical research focuses on international migration and urban settings, leaving rural-to-rural internal migration comparatively neglected. Given that rural areas in the Global South often bear the brunt of climate impacts and host large numbers of displaced people, there is an urgent need to understand how these communities respond to environmental migrants. This focus is especially salient because environmental migration is predominantly internal and often short-distance, frequently rural-to-rural[2,7,21]. Micro-level studies

[1]Department of Politics and Public Administration, University of Konstanz, Konstanz, Germany. [2]Department of Political Science, University of Zurich, Zurich, Switzerland. [3]Department of Humanities, Social and Political Sciences, ETH Zurich, Zurich, Switzerland. ✉e-mail: lukas.rudolph@uni-konstanz.de

corroborate this pattern specifically for Bangladesh, where actual environmentally induced moves are primarily rural-to-rural[3,22], and households facing erosion stress disproportionately aspire to relocate to nearby rural localities rather than urban centers[22,23]. These findings suggest that environmental mobility does not conform to a simple rural-urban transition; rather, it reflects strategies to minimize disruption by remaining in familiar settings, including sustaining agrarian livelihoods and norms, leveraging transferable skills for local work, and relying on kinship and social networks that structure settlement, thereby reducing cultural distance. Against this backdrop, examining attitudes in rural receiving areas is crucial, as these communities are likely to remain primary destinations for environmental migrants in Bangladesh and many similarly affected contexts in the Global South for internal, short-distance moves.

This paper aims to address this gap by examining host community attitudes toward internal environmental migrants in rural regions of Bangladesh. We develop a conceptual framework centered on two key mechanisms through which attitudes are shaped: (1) perceived deservingness, and (2) empathy rooted in social, spatial, and experiential proximity. In addition, we consider interpersonal contact as a more tentative, exploratory extension, given that our survey captures only coarse proxies for exposure and general ties and does not isolate contact with environmental migrants specifically. Each mechanism captures a distinct form of psychological and social proximity that influences how host populations evaluate newcomers. Rather than assuming that these mechanisms are mutually reinforcing, we treat them as analytically distinct and state the conditions under which they interact.

The first mechanism, perceived deservingness, draws on a substantial body of literature in social psychology and public opinion that explores how host populations morally evaluate different types of migrants. Deservingness refers to the belief that certain individuals or groups are more entitled to support or protection based on the involuntariness of their displacement[14,24]. Migrants forced to flee due to war, persecution, or environmental disasters are generally viewed more sympathetically than those who migrate for economic or personal reasons[25]. This distinction operates through what is often referred to as the deservingness heuristic, a cognitive shortcut that individuals use to determine who is worthy of empathy and support[24,26]. Empirical studies in the Global North have consistently found that environmental migrants are perceived as more deserving than economic migrants. For the US and German context, citizens seemingly extend more favorable attitudes toward individuals displaced by environmental events than those seeking work or higher income, though less favorable than toward political refugees[15]. Similar findings have emerged in Denmark[17] and again in Germany[16], suggesting a degree of cross-national consistency in how forced displacement is morally coded. These studies underscore the importance of attributional judgments, that is, whether the migrant is seen as responsible for their condition or as a victim of uncontrollable external forces in shaping public support. Yet, it remains unclear whether these dynamics hold in the Global South, where displacement is more commonly internal, and climate change interacts with chronic socioeconomic vulnerability. In such contexts, the distinction between forced and voluntary migration is often blurred[9], and the applicability of the deservingness heuristic may be conditioned by local norms, moral frameworks, and lived experiences of hardship. For example, urban residents in Kenya and Vietnam have been shown to not systematically differentiate between climate and economic migrants[11], suggesting that attributions of deservingness may not be universally salient. Our study engages this ambiguity directly by testing whether rural hosts perceive environmental migrants as more deserving than other groups and whether this perception translates into greater acceptance. In our setting, involuntariness is visible in daily risks, such as erosion, flood, and crop loss, not just a rhetorical label. Environmental migration is widely understood as involuntary, given strong place attachment, rather than

chosen, and hence we expect environmental migrants to be more welcome than economically motivated migrants (H1).

The second mechanism centers on empathy via social and geographic proximity. We conceptualize empathy as arising from perceived similarity. People tend to be more welcoming toward those who share their ethnic, linguistic, or religious background, and research on ethno-cultural threat shows that migrants from a perceived out-group may be seen as disrupting social cohesion[27–29]. Geographic proximity likewise conveys familiarity with migrants from nearby places more likely to share customs, networks, and environmental exposure, while those from more distant regions are more easily perceived as outsiders[27–30]. Emerging empirical evidence shows that individuals living closer to areas affected by environmental disasters are more likely to express solidarity with displaced people[31,32]. Taken together, these similarity cues, i.e., same religion and spatial proximity, shape whether migrants are seen as 'like us' and therefore worthy of inclusion, in line with the in-group/out-group logic of intergroup empathy[26,33]. However, separate from empathy, hosts also make instrumental judgments about the likely balance of contributions and demands newcomers bring. Occupational status is a key signal since higher-status occupations connote self-reliance and potential contribution, while low socioeconomic status is often associated with fears of strain on public finances and competition in the labor market[13,26]. This economic logic has been repeatedly documented in studies of migration attitudes and is particularly salient in resource-constrained settings, where practical concerns may temper otherwise sympathetic responses[34]. Accordingly, we expect independent main effects of religion, occupational status, and distance on acceptance (H2).

Beyond these independent effects, deservingness should do more than lift average support for involuntary migrants. In particular, by lowering blame, widening moral concern, and increasing the willingness to extend help across group boundaries, it should soften exclusionary responses to otherwise disfavored attributes[11,14,24,25]. Consequently, even when migrants differ in religion, originate from more distant areas, or hold low-status occupations, they may be judged less harshly if their displacement is clearly beyond their control. We therefore expect the environmental-motive (deservingness) cue to operate as a moderating force, attenuating the penalties associated with social and economic dissimilarity (H3).

In addition, because the salience of involuntariness most likely differs across respondents, the assessment of environmental migrants should vary with hosts' own exposure to environmental risk and mobility. Experiential proximity, i.e., having faced similar environmental hardship, provides a particularly strong basis for identification and compassion. Research shows that perceived similarity in hardship strengthens prosocial attitudes toward migrants[35], and that shared experiences of adversity can foster prosocial behavior and strengthen feelings of mutual responsibility[36]. Individuals with personal migration histories are also more supportive of those displaced by environmental events[32,37]. In our framework, we expect that rural host communities will respond more positively to environmental migrants, particularly when local residents have experienced environmental hardship and environmental migration[37]. By highlighting shared risks and experiences, experiential proximity can enhance host receptivity through mechanisms of perceived similarity and common fate. Experiential proximity therefore makes need more salient and invites perspective-taking, thereby strengthening the premium extended to involuntarily displaced migrants; correspondingly, the moral claim of need can attenuate negative reactions associated with different religion or distant origin, and temper reservations about low occupational status[11,14,24,25]. From an environmental-psychology perspective, affective reactions, especially climate worry or concern, predict support for climate policies[38]. Empathic concern is especially likely when loss and vulnerability are rendered vivid through shared hazard exposure or narratives that personalize displacement[39]. By making need salient and inviting perspective-taking, experiential proximity strengthens support for involuntarily displaced migrants; we therefore expect the deservingness effect to be strongest among respondents with greater experiential proximity (H4).

Third, as an extension, we also consider a conceptually distinct potential third mechanism, namely interpersonal contact with (environmental) migrants, which we can only test tentatively. This mechanism draws on intergroup contact theory, which posits that interpersonal contact can reduce prejudice and promote more inclusive attitudes[40,41], and empirical evidence from the Global North indicates that experience of contact with migrants is a predictor of more favorable attitudes[42]. Applied to the context of environmental migration, hosts who live in communities with recent in-migration and/or who have interacted with environmental migrants may better understand newcomers' challenges and be more willing to support their inclusion. At the same time, contact, and especially exposure that does not translate into frequent or meaningful interpersonal interaction, may be insufficient to shape attitudes[43,44], and can also backfire by heightening perceptions of competition or burden, particularly when resources are constrained[13,25].

In cases of ongoing environmental changes, we posit this mechanism as only partly testable, however: When rural-to-rural internal (environmental) migration is widespread, researchers observe little to no actual variation regarding the extensive margin of contact (no vs. any contact with (environmental) migrants). This is especially the case where severe (environmental) changes are ongoing over longer time spans, as in our case. Then, most localities will have received some extent of (environmental) in-migration. What we can observe, are differences in perceptions of actual in-migration, and self-reported intensity of contact (e.g., inclusion of migrants in friendship networks). Both have been shown to shape attitudes in past research[43,44]. We, therefore, assess whether residents in host communities with heightened perceptions of in-migration express more (or less) favorable attitudes toward environmental migrants. We do so for a context where permanent migration due to environmental factors is highly relevant[45], and, to a large extent, rural-to-rural[3,23]. This allows us to, tentatively, evaluate whether lived experience with environmental in-migration fosters empathy or amplifies perceived threat. Conceptually, the contact mechanism complements deservingness and social, spatial, and experiential proximity by highlighting that judgments reflect both perceptions of migrants and hosts' lived experiences. In the environmental case, contact can make the involuntariness of displacement more salient and credible, for instance, through observable loss and first-hand narratives, which may strengthen or, under scarcity, weaken the weight given to the environmental-motive signal (H5; exploratory).

Based on a conjoint experimental study in rural communities along the Jamuna River in Bangladesh, we investigate the above-mentioned mechanisms. We find that migrants displaced by riverbank erosion are significantly more likely to be accepted than economic migrants, consistent with a strong role for deservingness in shaping attitudes. Environmental displacement also appears to attenuate penalties tied to other migrant attributes, such as religion, occupation, and distance of migrant origin. Respondents who experienced similar environmental hardships, such as erosion-induced house loss, express the strongest preferences for environmentally displaced migrants, tentatively supporting the empathy mechanism. By contrast, using a coarse proxy for perceived exposure to in-migration, we find no measurable association with attitudes. Note, however, that this proxy offers limited leverage for adjudicating contact theory in a setting where rural-to-rural in-migration is long-standing, as it captures perceived in-migration rather than the frequency and quality of interpersonal interaction, and it does not isolate contact with environmental migrants specifically.

Our study makes three key contributions to the literature on migration and climate adaptation. First, it extends the analysis of migration attitudes to include environmental migrants, a group that remains underrepresented in public opinion research. Second, it shifts the empirical focus to internal, rural-to-rural migration in the Global South, where much climate-induced mobility actually occurs and where host perspectives remain poorly understood. We emphasize rural destinations because prior rural residence fosters familiarity with local socio-ecological settings, reducing cultural

distance despite resource constraints. Testing for a deservingness premium and its amplification by proximity in this context would provide strong evidence for the proposed mechanisms. Third, it develops and empirically tests a conceptual framework that integrates deservingness and empathy as core mechanisms, and contact as a more tentative, exploratory extension (given measurement limitations)—capturing distinct yet interrelated dimensions of how host communities respond to incoming migrants. By synthesizing insights from public opinion research, social psychology, and environmental studies, we provide a multidimensional account of the attitudinal drivers of migrant receptivity. The framework developed here allows for a nuanced understanding of when and why rural host communities may be more or less willing to accommodate environmentally displaced individuals. This has significant implications for policy, as host attitudes can facilitate or hinder adaptation efforts, shape the success of resettlement programs, and influence the prospects for social cohesion in regions at the frontlines of climate change.

## Results
### Case and study setting
Our study is situated in rural areas of Bangladesh along the Jamuna River (see Methods section 'Case' for details). While Bangladesh is highly vulnerable to extreme climate events[46], riverbank erosion is a prominent threat to livelihoods along the Jamuna, with an estimated 200,000 people displaced annually[47], and rural-to-rural migration is a common adaptation strategy[3,23].

In this context, we surveyed 265 respondents in migrant-receiving communities, ensuring that our inquiry into host community attitudes was realistic and relatable. A total of 156 respondents completed a binary profile conjoint experiment with three tasks ($N_{conjoint}$ = 2 profiles * 3 tasks * 156 respondents = 936), a well-suited method for investigating host attitudes toward migrants[14,34,48]. Conjoint experiments are relatively robust to social desirability bias[49] and demand effects[50], and have been shown to generalize well to real-world population preferences on migrant acceptance[51]. We presented the conjoint attribute levels visually, given a high share of illiterate respondents. In addition to the conjoint, respondents provided information on socio-demographics as well as migration experience and attitudes (see Methods).

### Population characteristics and descriptive results
We begin our presentation of the results by describing key characteristics of the study population, with a focus on attitudes toward migrants. A further summary of sample characteristics is reported in Supplementary Table A.1.

Using a context-adapted battery of standard migrant attitude items[43,52], measured on a five-point Likert scale ranging from strong agreement (1) to strong disagreement (5), the data suggest generally favorable baseline attitudes toward migrants (see Supplementary Table B.1). Respondents report low levels of job-related fear (mean = 3.9, model 2, constant), strong support for the deservingness of migrants (mean = 1.9, model 3, constant), and high agreement with the idea that migrants have a 'right to settle' (mean = 1.8, model 4, constant). Perceptions of economic motives as the primary driver of migration are moderate (mean = 3.0, model 5, constant), while assessments of similarity between migrants and the native population are relatively high (mean = 3.7, model 6, constant). In addition, respondents report no specific grievances with migrants (see Supplementary Table B.2). These results indicate a generally sympathetic stance toward migrants, despite the study being situated in a context marked by intense competition over natural resources, particularly land. Potentially, this is due to the fact that many in this context have their own migration history, also relating to environmental affectedness—only 45% of respondents indicate they have been living in the village since birth/marriage, and 56% indicate they have previously lost their home to erosion (see Supplementary Table A.1).

Hence, when asked directly (as shown in Supplementary Table B.3), respondents also express relatively high levels of migrant acceptance: an average score of 4 on a five-point acceptance scale. Notably, experimentally varying the question wording by framing migrant neighbors as individuals who might marry into the respondent's family, or whose children would

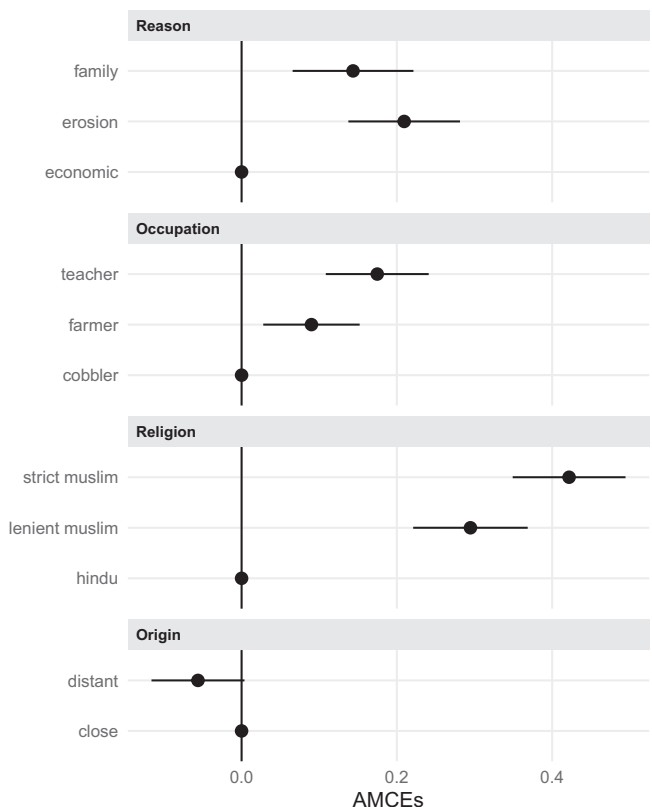

**Fig. 1 | Average marginal component effects (AMCEs) for the conjoint experiment.** Error bars indicate 95% confidence intervals from respondent-clustered standard errors ($N = 936$). Numerical results are presented in Supplementary Table B.4, model 1.

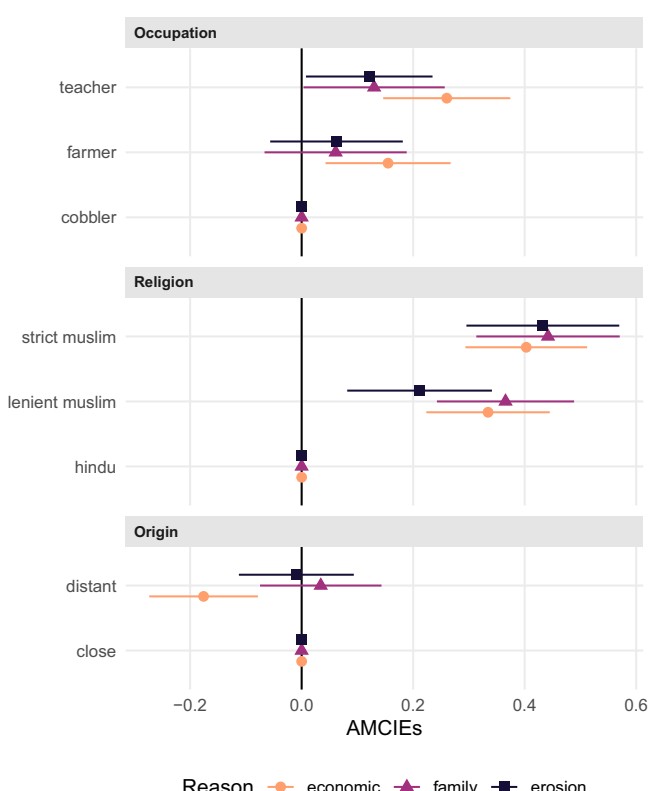

**Fig. 2 | Average marginal component interaction effects (AMCIEs) for occupation, religion, and origin attributes by reason of migration.** Orange circles denote AMCIEs from interactions with *economic*, purple triangles from interactions with *family* and dark purple squares from interactions with *erosion*. Supplementary Table B.5 presents the corresponding numerical AMCIEs. Error bars indicate 95% confidence intervals from respondent-clustered standard errors ($N = 936$).

attend school with the respondent's children, does not significantly alter this level of acceptance. These pre-registered wording treatments (see also Methods section 'Additional priming experiment') were designed to emphasize the potential personal impact of in-migration on respondents' daily lives. The consistently high baseline support, coupled with the insignificant wording treatment effects, suggests a generally welcoming environment toward migrants.

Taken together, these findings point to a local environment where additional immigration is not perceived as inherently problematic. This raises the next key question: In principle, which types of in-migrants are relatively more preferred by respondents?

## Main effects and attribute interaction effects from the visual conjoint experiment

To address this question, we turn to our results from the conjoint experiment. Figure 1 presents average marginal component effects (AMCEs) (for the corresponding numerical results, see Supplementary Table B.4 (model 1)). Compared to economic migration, migrant profiles citing family reunification are 14%-points more likely to be selected, while those citing erosion as the reason for migration are 21%-points more likely to be chosen. Both effects are not only statistically significant but also substantively meaningful. The difference between erosion and family reunification as migration motive (7%-points) is also notable, although the estimate lacks sufficient precision to reach statistical significance at conventional levels. Nonetheless, these findings already suggest a clear hierarchy in perceived legitimacy of migration motives: environmental displacement is preferred over economic migration (comp. H1).

The effects of the remaining attributes align with expectations: Migrants in higher-status occupations such as teachers (plus 18%-points) and farmers (plus 9%-points) are more likely to be selected than those in

low-status occupations (cobbler, serving as the reference category). Religious affiliation also plays a significant role. Muslim migrants are considerably more likely to be selected compared to Hindu migrants, the religious minority in the study area. Migrants identifying with a strict interpretation of Islam (plus 42%-points) receive the highest favorability, followed by those with a more lenient interpretation (plus 30%-points). Finally, long-distance migrants face a modest penalty of 6%-points ($p = 0.066$), but the estimate is imprecise and does not meet conventional significance thresholds. Hence, we find clear statistical support for main effects of religion and occupational status on acceptance, but not long-distance migration (comp. H2).

As indicated by the relative magnitudes of the AMCEs, respondents assign the greatest importance to migrants' religion when making their choices. The reason for migration ranks as the second most influential factor, followed by occupation and place of origin.

Next, we explore interaction effects by estimating Average Marginal Component Interaction Effects (AMCIEs), focusing on interactions between our core attribute of interest, reason for migrating, with each of the three remaining attributes: religion, occupation, and origin. Our aim is to assess whether environmentally motivated migrants face less discrimination based on other characteristics. Specifically, we examine whether the distribution of AMCEs is flatter when erosion is cited as the reason for migration, suggesting reduced sensitivity to other potentially stigmatizing attributes. In other words, the negative effects of less-preferred attributes (i.e., low-status occupation or minority religion) may be attenuated for environmental migrants, signaling a buffering effect of perceived deservingness. Results are presented in Fig. 2, estimated and presented via effects for (randomized) subgroups of reason.

Our expectation is largely supported by the data. The preference patterns for distance, religion, and occupation observed in Fig. 1 are fully replicated in the AMCIEs of economic migrants. However, for environmental migrants, these effects are notably attenuated. We continue by comparing environmental (dark purple squares, i.e., erosion-induced migrants) and economic (orange circles) migrants in detail.

The effect of occupation is substantively reduced: while economic migrants with a mid-status occupation (farmer) see relevantly higher choice probabilities compared to a low-status occupation (cobbler), by about 16%-points, this penalty shrinks to a substantively small 6%-points which is not statistically significant for environmental migrants. Similarly, the selection of migrants with a high-status occupation (teacher), which is very pronounced for economic migrants (choice probabilities increased by 26%-points relative to cobblers), is halved (to 12%-points) when erosion is the reason for migration. This difference in penalty sizes by economic vs. environmental migrant is estimated at $p = 0.095$ for teacher vs. cobbler, and even more imprecisely estimated for farmer vs. cobbler ($p = 0.294$) (see Supplementary Table B.6).

Religious identity follows a comparable pattern regarding migrants with lenient, but not strict, Muslim faith (compared to Hindu migrants). Strict Muslims see higher choice probabilities than Hindus, regardless of migration motive (increase by about 40%-points). In turn, migrant profiles including lenient Muslims are associated with a relevantly smaller advantage compared to Hindus when erosion is the reason for migration (21%-points) compared to economic reason of migration (33%-points). While substantively non-trivial, the estimate is imprecise ($p = 0.143$), and we cannot rule out no moderation.

Finally, distance to origin ceases to influence selection, suggesting that spatial proximity is no longer a relevant criterion when respondents consider migrants displaced by erosion. Specifically, economic migrants with distant origin see a relevant penalty (by 18%-points), which is reduced to nearly zero for environmental migrants. This difference in penalty sizes between economic vs. environmental migrant is statistically significant ($p = 0.025$).

Taken together, the overall pattern indicates that environmental migrants are evaluated more leniently across several social dimensions. This supports our hypothesis that perceived deservingness tempers exclusionary attitudes, with, potentially, the intriguing exception of strong religious faith (comp. H3).

Last, we interpret AMCIEs with family reunification as the reason for migration (orange triangles in Fig. 2). It is noteworthy that the patterns observed for erosion and family reunification as reasons for migration are remarkably similar. In both cases, respondents appear more accepting of otherwise less-favorable attributes such as low-status occupation or distant origin, compared to profiles citing economic reasons. This suggests that erosion and family reunification are perceived as similarly legitimate or deserving motives for migration, both likely triggering greater empathy and reducing discriminatory responses, consistent with theories of deservingness in public opinion research[14,26].

Supplementary Fig. B.1 presents marginal means[53], confirming that these interpretations are robust to the choice of reference category. In other words, the observed effects do not hinge on how the baseline is defined in the statistical model.

## Conjoint marginal means by respondent subgroups
Next, we examine heterogeneity in preferences across specific respondent subgroups. In particular, we assess whether responses to the conjoint experiment differ based on (a) localized personal experience with erosion risk, (b) perceived in-migration into the village, (c) the respondent's own migration history, and (d) whether respondents have themselves been displaced by erosion.

We use marginal means for comparisons across subgroups. Notably, we find no significant differences in marginal means based on whether respondents perceive riverbank erosion as the main environmental event (see Fig. 3, panel (a)); whether they report the recent arrival of new families

in their village (see Fig. 3, panel (b)); or whether they have migrated themselves (see Fig. 4, panel (a)). However, we observe meaningful heterogeneity among respondents who have personally experienced house loss due to riverbank erosion, that is, those with direct cognitive and emotional proximity to forced environmental displacement (see Fig. 4, panel (b)). This group shows a markedly stronger preference for environmentally displaced compared to economic migrants: respondents with their own grave erosion experience select erosion-based migrant profiles 62% of the time, compared to 38% for economic migration. Among respondents without such experience, erosion-based migrants are also preferred over economically motivated ones, but to a much lesser extent (54% versus 43%). The spread in marginal means, i.e., the increase in choice probabilities for environmental vs. economic migrants, amounts to 24%-points with, but only 11%-points without own erosion experience. The substantive size of this difference is notable and consistent with theories of experiential proximity and empathetic concerns—prior research has shown that personal exposure to hardship can foster empathy, particularly toward others perceived to share similar experiences[28]. However, in our case, the difference in marginal means across subgroups is estimated outside conventional levels of statistical significance ($p = 0.122$; see Supplementary Table B.8, model 4), calling for renewed tests whether, also in the context of migration, such experiential proximity may reduce perceived social distance and strengthen the sense of shared vulnerability, thereby increasing support for environmentally displaced individuals.

Note also that family reunification is evaluated equally in both groups (with choice probabilities of around 51% and 54%), indicating that past erosion experience induces differentiation, particularly among economic and environmental migrants. Taken together, these heterogeneity patterns are consistent with the idea that shared hazard exposure heightens the salience of involuntariness, thereby strengthening the deservingness premium for environmental displacement relative to economic motives (comp. H4), though insignificantly estimated at conventional levels. At the same time, we find no heterogeneity by extent of prior contact with migrants (comp. H5).

Lastly, in estimating the main AMCEs, the AMCIEs by reason for migration, and the differences between them, as well as marginal means and subgroup differences, we conduct a multitude of statistical tests. We therefore correct for multiple testing using the Benjamini-Hochberg procedure[54]. The results of these corrections are summarized in Supplementary Table B.10. In short, most of the main interpretations regarding H1 and H2 hold when accepting a False Discovery Rate of 0.05, and a False Discovery Rate of 0.1 regarding H3. However, for our direct evidence regarding H4, experiential proximity, we would have to accept a lenient False Discovery Rate of 0.2 (for details, see extensive interpretation in Methods section 'Corrections for multiple testing').

## Generalizability of main effects to the broader study area population and beyond
Finally, we assess whether our findings are likely to generalize to the broader population in the study area. Specifically, we examine to what extent our sample, comprising neighbors of actual in-migrants, is representative of the wider population living in the area. Given the possibility that migrants may purposefully select destinations, the characteristics of their neighbors may not reflect those of the general population. We, therefore, reweigh our sample to conform to a geo-spatial population-representative draw of the rural Jamuna population (see Methods section 'Entropy balancing for generalizability'). Re-estimating the AMCEs (Supplementary Fig. B.2) and AMCIEs (Supplementary Fig. B.3) using the reweighted data yields results that are substantively consistent with our main findings. The patterns of effect sizes remain nearly identical, and the interaction effects described in Results subsection 'Main effects and attribute interaction effects from the visual conjoint experiment' mostly reach conventional levels of statistical significance. We note, however, that standard errors are slightly larger, as expected, given that standard errors increase slightly due to the downweighting of some observations.

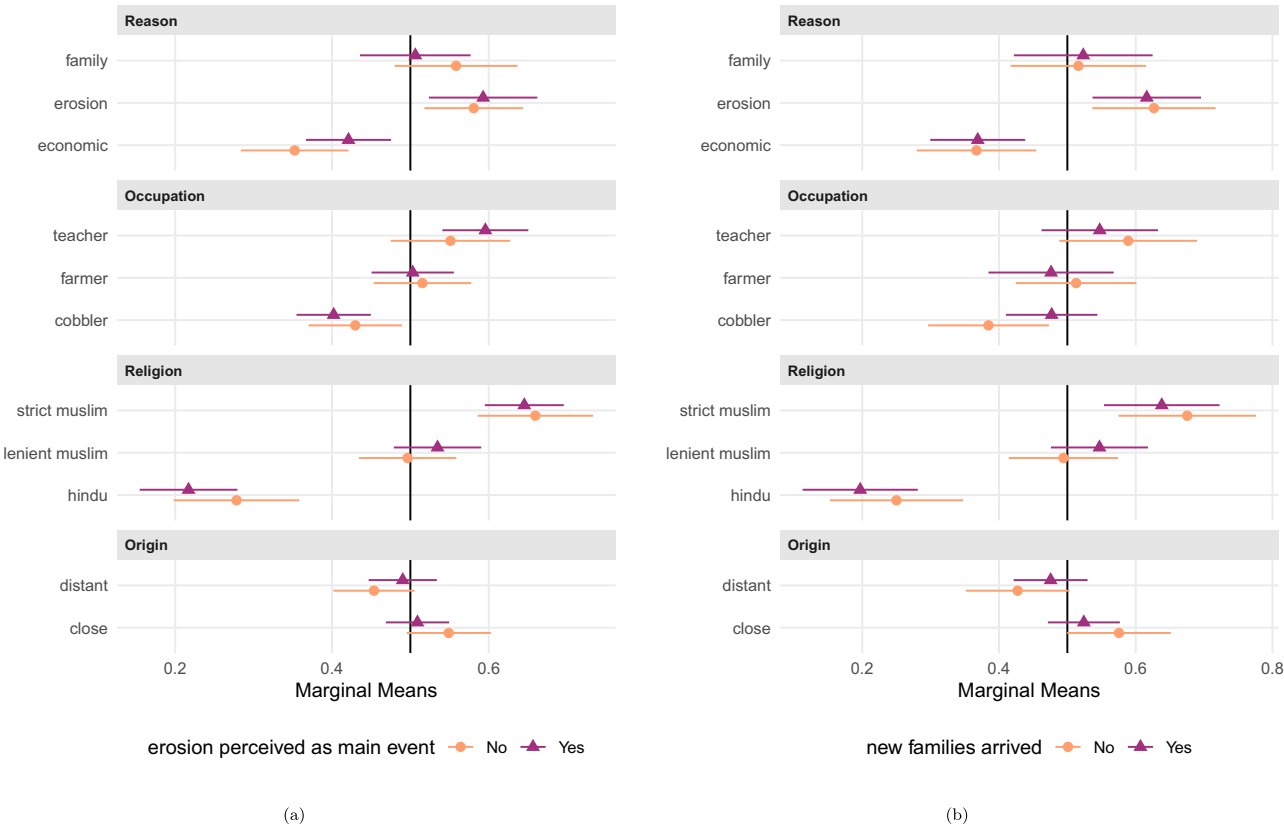

(a)　　　　　　　　　　　　　　　　　　　　　(b)

**Fig. 3 | Marginal means for conjoint experiment by subgroups of respondents (perception of erosion as main environmental stress; perception of in-migration).** Panel (**a**) shows marginal means for respondents perceiving (*N* = 534; purple triangles) vs. not perceiving (*N* = 378; orange circles) riverbank erosion as core environmental event in their village, and panel (**b**) marginal means for respondents

stating that new families (*N* = 288; purple triangles) vs. no new families (*N* = 216; orange circles) arrived in their village. Error bars indicate 95% confidence intervals from respondent-clustered standard errors. Numerical results are presented in Supplementary Table B.7, model 1 to model 4.

Overall, these robustness checks provide reassurance that our results are not driven by idiosyncratic features of our sample and are likely to hold across a broader population in the region. Of course, this does not imply that our findings necessarily generalize to other rural contexts in the Global South. Our study population is distinctive in combining high spatial and experiential proximity with generally low levels of migrant hostility (see Results subsection 'Population characteristics and descriptive results'). Such comparatively low migrant hostility also appears in Bangladeshi samples in cross-country data[55]. Hence, our context may be specific along multiple dimensions. This context specificity is, however, consistent with the broader literature, which reports that the perceived deservingness of climate-displaced migrants appears in Global North settings (e.g., in the US and Germany[15,16]), but appears weaker in some urban Global South environments[11]. Notably, these country cases display similar average acceptance levels[55], underscoring the role of additional moderating factors, such as spatial and experiential proximity, that we emphasize here. Future research could use our design as a template to further probe this heterogeneity across settings. In addition, in our case, differentiating respondents by their general acceptance of migrants does not substantially change favorability toward reasons for migration (see Supplementary Fig. B.4).

## Discussion

In this article, we explore how rural host communities in a climate-change-affected region of the Global South respond to internal environmental migrants. Drawing on theories of deservingness, empathy through geographic and experiential proximity, and treating intergroup contact as a more tentative, exploratory extension given measurement limitations, we examined how these mechanisms shape attitudes toward migrants displaced by environmental hazards, with a specific focus on

riverbank erosion in Bangladesh. Our findings reveal several key insights that contribute to both the scholarly literature and ongoing policy debates on climate-related mobility and social cohesion.

We find strong evidence supporting the relevance of perceived deservingness in shaping host attitudes toward internal environmental migrants (comp. H1). Respondents expressed clear preferences for migrants displaced due to environmental hazards, namely riverbank erosion, over those migrating for economic reasons. This effect is both statistically significant and substantively large: migrants citing riverbank erosion as their motive were 21%-points more likely to be selected as preferred neighbors compared to economically motivated migrants. This mirrors patterns observed in high-income countries[15–17], suggesting that the deservingness heuristic, that is, involuntary displacement triggering greater sympathy, extends to low-income, climate-vulnerable rural settings. Furthermore, the observed hierarchy of migration motives, where environmental and family-reunification migration are both preferred over economic migration, indicates that hosts make meaningful moral distinctions between different reasons for mobility and that broader deservingness considerations are not exclusive to wealthier societies but may also operate in diverse cultural and socioeconomic contexts.

However, our results diverge from those of Spilker et al.[11], who found no such deservingness effect in the context of rural-to-urban migration in Kenya and Vietnam. In their study, urban residents did not systematically favor environmental migrants over economic ones, indicating that environmental displacement did not confer greater perceived legitimacy or moral worth. This contrast suggests that the salience of the deservingness heuristic may be dependent on contextual factors that remain insufficiently explored. In rural-to-rural settings like ours, where environmental hazards are immediate, visible, and deeply entwined with local livelihoods, hosts are

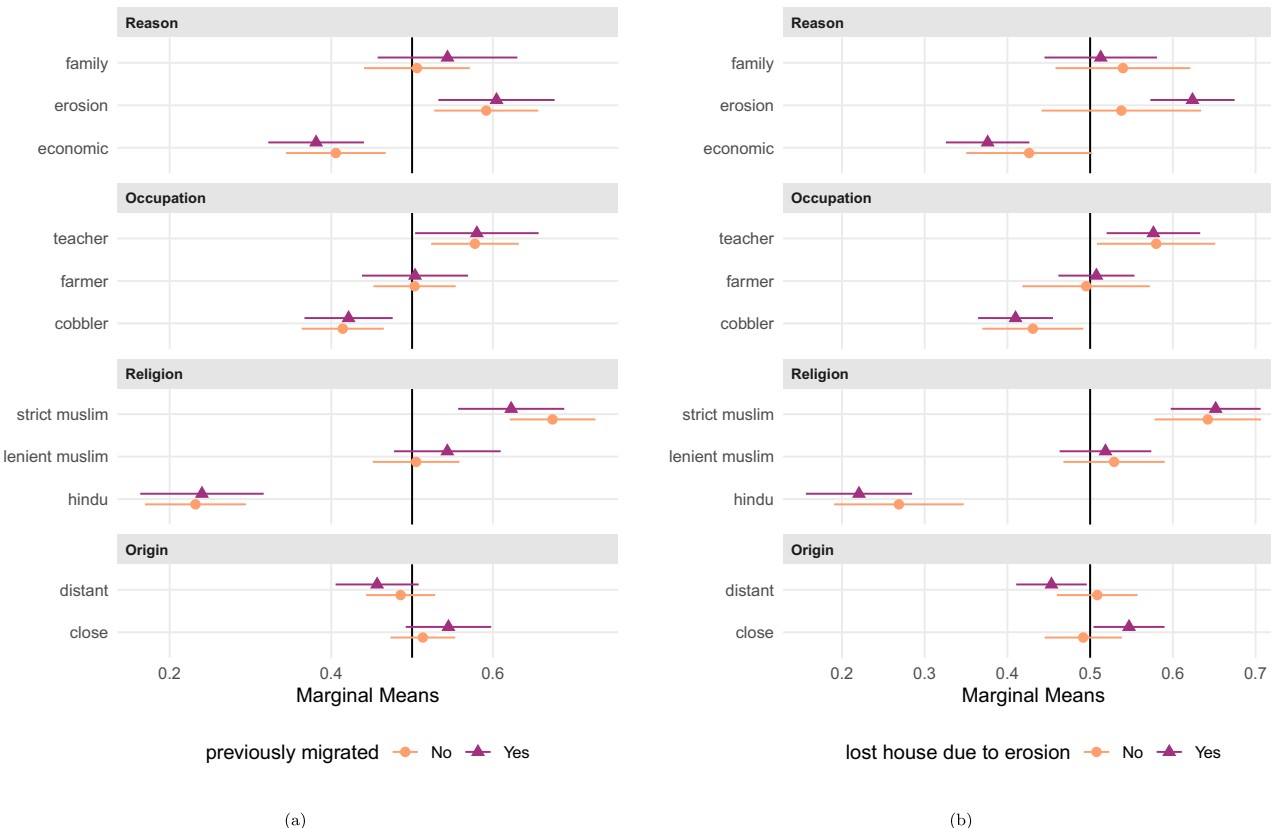

**Fig. 4 | Marginal means for conjoint experiment by subgroups of respondents (previous own migration experience; previous own erosion affectedness).** Panel (**a**) shows marginal means for respondents that previously migrated to current location (N = 408; purple triangles) vs. those that did not (N = 510; orange circles) and panel (**b**) marginal means for respondents stating that they have lost their house

(N = 576; purple triangles) vs. not lost their house (N = 354; orange circles) due to erosion at some point in the past. Error bars indicate 95% confidence intervals from respondent-clustered standard errors. Numerical results are presented in Supplementary Table B.7, model 5 to model 8.

more likely to interpret environmentally-induced migration as a credible and morally compelling form of forced displacement. In contrast, urban hosts may struggle to disentangle environmental and economic motives—particularly when rural in-migration is common and often associated with labor mobility. As a result, environmental migration in urban contexts may be perceived as more voluntary or opportunistic, weakening the cognitive basis for deservingness-based evaluations. This divergence may also reflect differences in how migrant narratives align with local expectations. In rural Bangladesh, environmental migrants often conform to a familiar and culturally resonant profile: landless households displaced by natural hazards, seeking shelter and solidarity among peers. These narratives may reinforce the moral intuition that such migrants are deserving of empathy and support. In urban areas, however, environmental migrants may be subsumed under broader stereotypes of rural job-seekers or slum-dwellers, diluting the force of environmental victimhood and triggering concerns about resource strain, informality, or social disorder. Hence, when these dominant frames prevail, deservingness cues may lose their power to shape attitudes.

Importantly, our findings also show that the perceived deservingness of environmental migrants moderates other exclusionary forms (comp. H3). We find clear evidence that environmental migrants faced less penalty for a characteristic that otherwise reduced support, being from a distant location (p = 0.025). This buffering effect suggests that when migrants are perceived as involuntarily displaced, host respondents become less sensitive to other potentially exclusionary attributes. Regarding low-status occupation (p = 0.095) and lenient faith expression (p = 0.143), two other characteristics that otherwise reduced support, estimates are not supporting, but indicative of a similarly lesser penalty for environmental migrants. In essence, we conclude that deservingness likely functions not only as an independent source of support but also as a moral filter that softens social boundaries.

These findings underscore the need to examine how host attitudes are shaped by interactions between migrant attributes and context-specific norms of legitimacy, rather than treating these factors as isolated dimensions. Most directly, our results suggest that spatial-proximity considerations operate in tandem with perceived deservingness: they matter more for economic migrants, who may be viewed as more voluntarily mobile, than for environmental migrants, whose movement may be interpreted as compelled by necessity.

In addition, our results shed light on the role of empathy grounded in social and geographic proximity in general (comp. H2), and on experiential proximity for environmental migrants in particular (comp. H4). Regarding social and geographic proximity, and averaging across migration motives, respondents showed a moderate preference for migrants from nearby regions (p = 0.066), although the estimate is modest and imprecise, and a clear preference for migrants sharing the dominant religion (p < 0.01), mirroring findings on general migrant acceptance[26]. Our findings resonate with those of Lujala et al.[32], who show that proximity to environmental risks enhances host support for environmental migrants. We build on this by investigating whether the effect of proximity could be conditional, becoming more salient when migration results from involuntary, environmental causes rather than economic motives. Regarding experiential proximity, we observed a substantively large, though insignificantly (p = 0.122) estimated positive coefficient: respondents who had themselves lost their homes due to erosion were considerably more supportive of erosion-affected migrants than those without such experience. While insignificantly estimated, the magnitude of effects (about 13%-points) aligns with theories suggesting that personal exposure to hardship fosters empathy and prosocial behavior[28,30]. Hence, our results could be indicative of a relationship between experiential proximity and acceptance. Given statistical imprecision, we cannot provide

conclusive evidence for the mechanism in this sample, however. Future work with larger samples is needed to assess whether experiential proximity systematically amplifies support for environmentally displaced migrants. Note that, in such endeavors, future research should also assess potential confounders more thoroughly. For example, experiencing environmental hardship may correlate with social class—and lower social class in itself appears to be correlated with prosocial behavior[56]. While our design is not tailored to causally differentiate respondents by the moderator environmental hardship, we consider major confounding unlikely. First, comparisons on observable markers of economic well-being, such as (objective and subjective) household income or education, seem largely comparable in our sample between individuals reporting environmental hardship and those who do not (see Supplementary Fig. B.5). Second, riverbank erosion affects longstanding communities in a quasi-random fashion, making economic well-being unlikely to be a strong predictor of affectedness[23].

Finally, our evidence on intergroup contact tentatively indicates limits to the usefulness of this analytical lens in settings of high-volume rural-to-rural environmental migration (comp. H5). Previous work in both high- and low-income settings has emphasized that personal contact with migrants can reduce prejudice[41,42]. In our context, however, all sampled villages were migrant-receiving, partly due to our sampling strategy, but also because permanent rural-to-rural migration has been pervasive in this setting for decades[3,23,45]. Against this backdrop, we find no credible evidence that perceived exposure to migrants in the village, i.e., differences between respondents who reported new families arriving in their village and those who did not, is associated with meaningful differences in migrant acceptance preferences. Although this conclusion is tentative given our crude measures of contact, we suspect it reflects the particular nature of rural-to-rural migration in this context, where high baseline familiarity and cultural homogeneity reduce the salience of intergroup boundaries and where such migration processes have been ongoing for a long time. It may also be the case that contact must be of a certain quality, for instance, frequent, cooperative, and perceived as positive, to shape attitudes, consistent with the conditions specified in classic contact theory[40,57]. Finally, when we extend the analysis to compare respondents who report having migrants as close friends (vs. not), we likewise find no credible evidence of differences in favorability by reason for migration (although a slightly lower religious bias emerges, see Supplementary Fig. B.6). While most actual permanent in-migration in this area is likely environmental, we acknowledge that general perceptions of in-migration are a crude measure for assessing how contact shapes the acceptability of migrants with environmental vs. economic motivations. Future research should investigate this more explicitly, for example, by inquiring about the extent of respondents' contact with migrants who moved for different motivations. This would be a fruitful extension, given that respondents appear to generalize from the motivations of migrants they encounter to group-level motivations, and in turn to attitudes toward these groups in other contexts[58].

The high baseline support for migrants across our sample, regardless of experimental framing or wording, also merits attention. Respondents reported low levels of concern about job competition, high agreement with migrants' right to settle, and a lack of perceived increase in intergroup conflict. These results may reflect a broader sociocultural ethos of hospitality and resilience in rural Bangladesh, or they may stem from the localized, kinship-based structure of migration in this region. Unlike urban settings, where migrants often arrive as strangers, rural migrants in our study were usually from nearby areas and often shared socio-economic and cultural backgrounds with their hosts. This social familiarity may foster a form of implicit contact, where shared identity and circumstance reduce the perceived social distance between hosts and migrants. At the same time, our results caution against assuming that rural host communities will always be welcoming or resilient to in-migration. The religious penalties observed in our conjoint experiment, particularly for Hindu migrants, highlight that ethnoreligious identity continues to shape attitudes, even in contexts where economic or environmental concerns are less pronounced. This underscores the need to consider intersecting dimensions of identity when designing policies aimed at promoting inclusion. The fact that religious bias is attenuated, but not erased, when migrants are perceived as deserving suggests that normative appeals to necessity or vulnerability may only partially mitigate deeper-seated prejudices.

From a methodological standpoint, our study demonstrates the utility of conjoint experiments for disentangling complex attitudinal mechanisms in low-literacy and rural environments. By adapting the conjoint design with verbal presentation and visual aids, we were able to generate robust causal estimates of attribute-level effects among a population that is often excluded from experimental research. Furthermore, our use of entropy balancing and robustness checks confirms that the results are not driven by sample selection, enhancing the generalizability of our findings to broader populations in climate-affected rural regions.

Together, these findings contribute to a growing literature on the social and political consequences of climate change, not only through the lens of migration drivers but also through the receptive capacities of host communities. As internal displacement due to climate hazards becomes increasingly common in the Global South, understanding host attitudes will be central to managing social cohesion, minimizing conflict, and designing equitable adaptation strategies. Our study offers an empirically grounded and theoretically informed account of how deservingness, empathy, and experience relate to these attitudes, and it provides actionable insights for both policymakers and practitioners seeking to foster inclusive and adaptive rural communities.

## Conclusion

This paper examines host community attitudes toward internal environmental migrants in rural Bangladesh, focusing on how perceptions of deservingness as well as geographic and experiential proximity shape receptivity, and providing an exploratory probe of exposure/contact using coarse proxy measures. We find strong support for the deservingness heuristic: migrants displaced by riverbank erosion are consistently preferred over those migrating for economic reasons ($p < 0.01$). Environmental displacement attenuates the distance penalty ($p = 0.025$), while estimated attenuation for occupation ($p = 0.095$) and religion ($p = 0.148$) is imprecise. Shared experience of climate-induced displacement (losing one's home due to erosion) is estimated to be positive, with higher acceptance of erosion-affected migrants, though the estimate is insignificant ($p = 0.122$). In contrast, using our coarse exposure/contact proxies, we find no credible evidence of heterogeneity in acceptance. While our evidence comes from rural Bangladesh, similar qualitative patterns are most likely where rural receiving communities face visible, involuntary environmental loss and relocation occurs nearby through kinship and shared livelihoods. In such settings, a deservingness premium should emerge and be amplified by proximity; where baseline acceptance is lower or cultural/spatial distance is greater, for example, urban or long-distance moves, magnitudes may attenuate or differ. For policy, this implies facilitating near-site relocation and family reunification, communicating the involuntary/need-based nature of displacement, and pairing newcomer support with host-community investments in services and livelihoods to ease resource competition.

## Methods

To evaluate our empirical expectations, we conducted a face-to-face survey with rural residents in northern Bangladesh ($N = 265$) from January to February 2024[59]. Respondents were not themselves recent migrants, to isolate the attitudes of host community members. At the core, the survey included a survey-embedded, binary profile, forced-choice conjoint survey experiment, successfully completed by 156 respondents over three rounds ($N_{conjoint}$ = 2 profiles * 3 tasks * 156 respondents = 936). Our study received ethical approval from the ETH Zurich Ethics Commission (Decision EK 2020-N-67). All respondents provided written informed consent prior to participating in the survey. The study was pre-registered[60]; deviations from the pre-registration are described in Supplementary Note C.1.

## Case

Due to its geographic features—including an extensive coastline and a dense network of rivers—Bangladesh is highly vulnerable to extreme climate events[46]. One such event is riverbank erosion, which is driven by intense rainfalls and frequent flooding. This phenomenon severely disrupts livelihoods by damaging agricultural land, homes, and infrastructure. As climate change progresses, the frequency and intensity of riverbank erosion is projected to rise, with an estimated 200,000 people displaced annually as a result[47]. Faced with the loss of land and livelihoods, affected populations turn to migration as an adaptive strategy, seeking to improve their long-term economic security and resilience[61,62]. Unlike the rural-to-urban patterns observed in other contexts, those displaced by riverbank erosion in Bangladesh often relocate to nearby rural areas, maintaining proximity to their home of origin[3,23]. This tendency is partly explained by the importance of kinship networks, which function as critical support systems during times of crisis[61,63]. Because riverbank erosion tends to generate localized, short-distance migration, host-migrant proximity—both spatial and social—is often high. In many cases, host communities either live in close geographic proximity to the migrants' areas of origin or have direct or indirect experience with erosion-related displacement themselves. This closeness increases the likelihood that host residents can empathize with the challenges faced by environmental migrants, whether through shared experiences or a strong sense of situational familiarity.

## Sampling strategy

The data analyzed in this study come from a survey conducted in rural Bangladesh containing standard survey questions (socio-demographics, past migration/environmental experiences, and migrant attitudes) and, at the core, a conjoint experiment. The survey was implemented as part of a broader research project (see https://data.snf.ch/grants/grant/185210) investigating environmental migration among communities on the eastern riverbank of the Jamuna River. The larger research project included a panel survey that tracked individuals who moved out from their original villages to their new destinations. After each (migrant) respondent was interviewed in their new location, survey enumerators identified and recruited nearby non-kin neighbors to participate in the conjoint experiment. Note that we targeted both households that neighbor cross-municipality (migration to another village) and within-village movements (movement within the same village, i.e., shifting the household). Neighbors were defined as households located in the immediate vicinity, i.e., the first row of houses surrounding the migrant respondent's house. To avoid bias due to kinship ties, we excluded relatives of migrant respondents from the sample, under the assumption that relatives would be more inclined to welcome migrants into their family network. For each migrant respondent, enumerators surveyed up to three neighboring respondents, depending on their availability. Eligibility for participation was limited to household heads (defined as the primary decision-maker within the household) or senior female members of the household. To caution against potential bias from sampling, we also implemented a pre-registered balancing strategy, indicating representativeness for the broad general rural population of the area—conditional on a broad set of observable respondent characteristics (see Results subsection 'Generalizability of main effects to the broader study area population and beyond').

The surveys were administered by a team of trained Bangladeshi enumerators, recruited from various universities across the country. Prior to the data collection, enumerators participated in intensive, multi-day in-person training sessions led by the authors, which covered the survey content, sampling procedures, ethical considerations, and use of the survey platform. Data were collected using the Qualtrics Offline Survey App, allowing for consistent administration across remote rural locations. The authors closely supervised fieldwork, monitoring the data collection process in real time, reviewing incoming data for errors, and providing daily feedback to enumerators to ensure data quality and consistency. As a token of appreciation for their time and participation in the survey, respondents received a modest monetary incentive.

## Conjoint design and estimation strategy

In the survey, respondents participated in a conjoint experiment[48], a widely adopted method in political science for examining public attitudes towards migrants in the last decade[14,34]. This approach allows researchers to isolate the causal effects of variation in multidimensional study objects, which regularly co-vary in real-world settings, on favorability—such as individual migrant attributes like occupation, religion, or reason for migration—by presenting respondents with hypothetical migrant profiles that closely resemble real-world cases. Among the most commonly studied attributes in these designs is the motive for migration, making conjoint experiments particularly well-suited for exploring attitudes towards climate-induced displacement. In addition, conjoint experiments have been shown to relevantly mitigate social desirability bias, which direct questions on migrant acceptance are likely prone to[64], by concealing both researcher objectives[50] and respondents' preferences on singular attributes[49]. Moreover, factorial survey experiments have been shown to capture actual population preferences well in our study context[65], and the binary-profile forced-choice design we employ has been shown to generalize well to actual population preferences on migrant acceptance[51]. As a result, several recent studies have employed this method to assess public attitudes toward climate migrants[11,15–17]. Our study builds on this growing body of work by adapting the conjoint design to the context of Bangladesh, a region experiencing both high climate vulnerability and significant internal migration.

Concretely, we fielded a forced-choice conjoint experiment, in which respondents were presented with pairs of (randomized) hypothetical migrant profiles and were asked to indicate which migrant they would prefer as a neighbor. Each profile was defined by four experimentally varied attributes: reason for migration, occupation, religion, and distance to origin. The levels for each attribute are listed in Fig. 5.

We selected these attributes based on their theoretical relevance to deservingness, social identity, and host-migrant boundaries in the Bangladeshi context. Regarding deservingness, the comparison of conjoint profiles that vary the reason for migration implicitly controls for two core alternative channels of migrant acceptance: economic competition (proxied by occupation) and cultural threat (proxied by religion and origin). We can therefore interpret the 21%-point higher choice probability for environmental relative to economic migrants as a preference structure net of these alternative channels. The underlying problem addressed here is so-called masking[66], whereby respondents infer multiple characteristics from a single attribute (i.e., not only deservingness). We believe that our design, which simultaneously fields these attributes, follows core explanations in the literature (economic; cultural/symbolic[26]), and similar conjoint studies have drawn comparable deservingness interpretations[14,15]. We concede, however, that favorability towards environmental relative to economic migrants could also relate to attributes we did not include.

Each migrant profile was independently randomized across these attributes, allowing for the identification of causal effects for each dimension. The order of attributes was randomized to caution against potential ordering bias[67].

Given the low literacy rates in the study region, the standard written format for conjoint experiments presenting the two migrant profiles with a tabular textual display was infeasible. Instead, survey enumerators told a standardized story, i.e., a verbal description of a hypothetical scenario to each respondent, outlining the four migrant attributes and their respective levels. To aid comprehension, a visual table displaying the attribute levels was shown in parallel to respondents on the enumerators' tablets or smartphones. Figure 5 displays the four attributes and their levels, together with the images used to visualize the different attribute levels. Supplementary Fig. A.1 provides an example of the table presented. Each respondent completed three paired profiles. Out of the 165 individuals who participated in the survey experiment, 156 were assessed by the enumerator as having understood the task. Restricting the analysis to these respondents yields a total of 936 recorded choices (respondents *

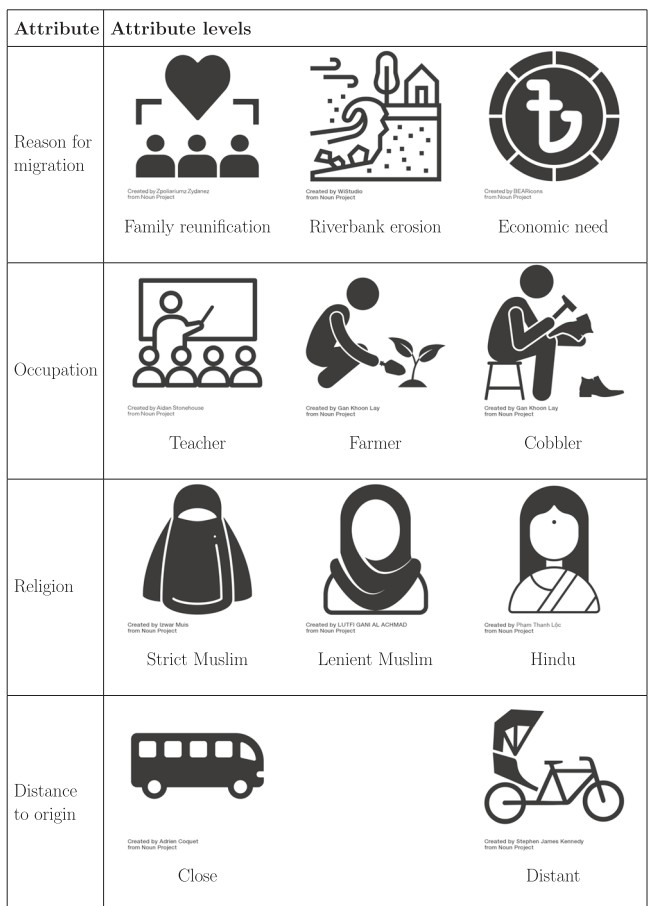

**Fig. 5 | Overview of attribute levels including exemplary pictorial representations of how attributes were shown to survey respondents on enumerators' smartphones/tablets.** For all but the religion attribute, levels were displayed with equal probability. For the religion attribute, Hindu was randomly displayed in 20% of cases, while strict Muslim and lenient Muslim were each displayed in 40% of cases. Note that, for reasons of publication under a CC-BY license, part of the icons are visually distinct but conceptually equivalent to what was shown to respondents (see Supplementary Note C.2 for details). All icons presented in Fig. 5 are licensed under CC-BY 3.0 from The Noun Project. Artist names are shown below the icons. Full credits, including links, are available in Supplementary Note C.2.

2 profiles * 3 tasks). Note the deviation between our overall study N of 265 and the conjoint N of 156 respondents, as technical issues with the survey app prevented conjoint data collection for the first interviews. This problem was also added as a note to the pre-registration[60]. Supplementary Table A.1 provides a comparison of summary statistics for the full sample and the sample of conjoint participants, indicating no relevant differences.

　　We report marginal means and average marginal component effects (AMCEs) of the respondents' choice, estimated using linear probability models[68]. The AMCE captures the effect of a specific attribute level on the probability that a migrant family is chosen, relative to the attribute's baseline level, averaged across all other attributes. This makes AMCEs suitable for causal interpretation of attribute-level effects. In addition to these main findings, we examine whether respondents' preferences vary depending on the migrant's reason for migrating. To explore this, we estimate Average Marginal Component Interaction Effects (AMCIEs), which capture inter-action effects between the reason for migration and the other migrant attributes.

　　Finally, to assess whether respondents' choices are shaped by their own experiences, we report marginal means and estimate treatment-by-covariate subgroup effects between the main outcome variable (respondents' choice) and four binary potential moderators:

- localized exposure to erosion risk. Question wording: "From your perspective, what were the main natural disasters that have happened in [village name] during the last 5 years?" Respondents could select multiple options. This variable is coded as 1 if 'erosion' was mentioned among the responses, and 0 if 'erosion' was not mentioned.
- perceived in-migration into the village. Question wording: "Do you know whether families have newly arrived in [village name] within the last year?" This variable is coded as 1 if the respondent answered 'yes', and as 0 if the respondent answered 'no'. Due to a survey filter, this question was asked of 46% of the sample, households likely to have a new in-migrant neighbor rather than a within-village mover.
- the respondents' own migration history. Question wording: "Has your household continuously been in [village name] since your birth (male) [females: 'since your marriage'] or have you moved here from another place?" This variable is coded as 1 if respondents indicate 'moved from another place', and as 0 if respondents (male) indicate 'since birth' [females: 'since birth' or 'since marriage'] otherwise.
- prior displacement due to erosion. Question wording: "Over your whole life, did riverbank erosion ever have an impact on your household?" If respondents answered affirmatively, they were asked: "What was the impact?" Respondents are coded as 1 if they reported an impact and that impact was the loss of a house, and as 0 if they did not report the loss of a house as an impact.

Descriptive statistics for these covariates are provided in Supplementary Table A.1.

### Additional priming experiment

Note that we also pre-registered a priming experiment[60], following the design strategy proposed by Alesina et al.[12]. Half of the sample was exposed to a series of questions designed to bring the issue of in-migration to their village to the forefront of their thinking before responding to the migrant attitude battery. However, unlike the findings reported by Alesina et al.[12], the priming intervention in our study had no measurable effect on response patterns. As shown in Supplementary Table B.1, the coefficient for the 'Migration prime group' is close to zero and statistically insignificant—both for the composite attitude index derived via principal component analysis (model 1), and for each of the individual components (models 2–6). This divergence is likely due to contextual differences, as discussed in the Results subsection 'Population characteristics and descriptive results': in our setting, migrants are generally viewed favorably and not perceived as posing an economic burden.

### Entropy balancing for generalizability

To address concerns regarding sample selection in our study population, we reweight our observations to conform to a population-representative draw. While up-to-date aggregate statistics for the socio-demographic composition of the region are lacking (the most recent census data date back to 2010), we can use a complementary dataset consisting of a geospatially randomized sample of communities residing along the Jamuna River[69]. This comparison dataset originates from the same research project and follows a pre-registered spatially clustered sampling strategy designed to ensure population-representativeness for rural communities residing along the Jamuna River[70]. This sample provides the best available approximation of general population characteristics in the region. Supplementary Table B.9 reveals some notable compositional differences between our sample and the broader population. In particular, our respondents tend to be younger and more likely to be female. It is worth noting, however, that other observable characteristics, such as household size, education levels, income, and housing conditions, are broadly similar between the two samples in simple bivariate comparison. Still, to account for these differences, we apply entropy balancing[71] to reweight our sample based on observable characteristics, including respondent demographics (age, marital status, gender, education, religion, household composition), income, and housing

conditions (e.g., building materials, access to sanitation, water, and electricity). The reweighting aligns the first, second, and third moments of the covariate distributions between our sample and the comparison population.

## Corrections for multiple testing

In order to correct for multiple testing, we make use of the Benjamini-Hochberg procedure and the concept of the False Discovery Rate[54], i.e., what is the probability that a $p$-value reported as 'significant' is actually a false-positive result? To do so, we collect the $p$-values for all the hypothesis tests reported in Figs. 1–4, which we present in the Results section. This concerns the $p$-values for: the seven estimates for the AMCEs for each conjoint level (Fig. 1 and numerical in model 1 of SI Table B.4, for H1 and H2); the five estimates for the differences between AMCIEs for the erosion vs. the economic level (Fig. 2 and numerical in model 1 of Supplementary Table B.6, for H3); and the four estimates for differences between marginal means for the erosion level between respondent subgroups (Figs. 3 and 4 and numerical in Supplementary Tables B.7 and B.8, for H4 and H5). The results of these corrections are summarized in Supplementary Table B.10.

We interpret Supplementary Table B.10 as follows: Overall, results show that our main deservingness finding is robust to corrections for multiple testing under strict FDRs of 0.05 and 0.1. The finding that environmental migrants receive a weaker penalty for a distant origin compared to economic migrants is robust under an FDR of 0.1. Two additional findings are only supported under a more lenient FDR of 0.2: first, that occupational penalties (teacher vs. cobbler) are attenuated for environmental relative to economic migrants; and second, that prior exposure to erosion (experiential proximity) moderates acceptance of environmental migrants. Finally, the attenuation of religious penalties (lenient Muslim vs. Hindu) for environmental versus economic migrants is supported only under an FDR of 0.3. The fact that these moderating effects hold only under relatively lenient FDR thresholds is consistent with the reported and substantively meaningful group difference estimates, which, however, mostly fail to reach conventional levels of statistical significance—potentially due to a limited sample size, but potentially also because these are actually false positive results. We therefore interpret these findings with caution, and call on future research to re-test these tentative findings in fresh samples with appropriate statistical power.

## Reporting summary

Further information on research design is available in the Nature Portfolio Reporting Summary linked to this article.

## Data availability

The data used in this study are available on Zoendo under accession code https://doi.org/10.5281/zenodo.18299881.

## Code availability

Replication code is available on Zonedo under accession code https://doi.org/10.5281/zenodo.18299881.

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

## Acknowledgements

We are grateful to two anonymous reviewers and the editors for their constructive feedback. Audiences at ETH Zurich, LMU Munich, and the University of Konstanz provided helpful comments. We thank the enumerator team in Bangladesh for their effort, and Stefan Wehrli and David Presberger for their support regarding the technical implementation of the conjoint experiment. We acknowledge financial support by Swiss National Science Foundation grant No. 185210 (PI: V.K.; Co-PI: L.R.).

## Author contributions

L.R., J.F., and V.K. conceived the study. J.F. and L.R. contributed to field work. J.F. and L.H. contributed to data preparation. L.H. and L.R. contributed to data analyses. V.K. (PI) and L.R. (Co-PI) received the study grant. All authors contributed to writing and revising the manuscript.

## Funding

## Competing interests

The authors declare no competing interests.
