## [Transparent Peer Review file · Communications Earth & Environment]

Empathy Informs Host Community Attitudes toward Climate Migrants in Rural Bangladesh

Corresponding Author: Dr Lukas Rudolph

Version 0:

Decision Letter:

Dear Dr Rudolph,

Your manuscript titled "'People Like Us': Empathy and Acceptance of Climate Migrants in Rural Bangladesh" has now been seen by 2 reviewers, and we include their comments at the end of this message. They find your work of interest, but some important points are raised. We are interested in the possibility of publishing your study in *Communications Earth & Environment*, but would like to consider your responses to these concerns and assess a revised manuscript before we make a final decision on publication.

We therefore invite you to revise and resubmit your manuscript, along with a point-by-point response that takes into account the points raised. Please highlight all changes in the manuscript text file. In particular, we wanted to flag Reviewer 1's final comment on direct measurements helping to strengthen support for causal claims. This is an important point, and one that we feel is particularly important that you address. In order for the study to be published in *Communications Earth & Environment*, your study should provide compelling novel insights on the acceptance of climate migrants in rural Bangladesh and the distinct role of each mechanistic pathway (deservingness, empathy, contact).

Please submit your point-by-point responses as a separate file, distinct from your cover letter where you can add responses to the Editors' comments that you do not want to be made available to the reviewers. Word files are preferred. We recommend that any figures, tables or graphs that are included in the response to reviewers are also included in the main article or Supplementary Information.

Please use the following link to submit your revised manuscript, point-by-point response to the referees' comments (which should be in a separate document to any cover letter), a tracked-changes version of the manuscript (as a PDF file) and the completed checklist:

Link Redacted

We hope to receive your revised paper within six weeks; please let us know if you aren't able to submit it within this time so that we can discuss how best to proceed. If we don't hear from you, and the revision process takes significantly longer, we may close your file. In this event, we will still be happy to reconsider your paper at a later date, as long as nothing similar has been accepted for publication at *Communications Earth & Environment* or published elsewhere in the meantime.

Please do not hesitate to contact us if you have any questions or would like to discuss these revisions further. We look forward to seeing the revised manuscript and thank you for the opportunity to review your work.

Best regards,

Niheer Dasandi, PhD

Editorial Board Member
Communications Earth & Environment
orcid.org/0000-0002-8708-837X

Yann Benetreau, PhD
Consulting Editor, Communications Earth & Environment
Deputy Editor, Communications Sustainability
Nature Portfolio
NY office
ORCID: 0000-0002-1897-0887

EDITORIAL POLICIES AND FORMATTING

- Behavioural and social science
- Ecological, evolutionary & environmental sciences
- Life sciences

Furthermore, please align your manuscript with our format requirements, which are summarized on the following checklist: <https://www.nature.com/documents/commsj-phys-style-formatting-checklist-article.pdf> Communications Earth & Environment formatting checklist

and also in our style and formatting guide <https://www.nature.com/documents/commsj-phys-style-formatting-guide-accept.pdf> Communications Earth & Environment formatting guide .

*** DATA: Communications Earth & Environment endorses the principles of the Enabling FAIR data project (<http://www.copdess.org/enabling-fair-data-project/>). We ask authors to make the data that support their conclusions available in permanent, publically accessible data repositories. (Please contact the editor if you are unable to make your data available).

All Communications Earth & Environment manuscripts must include a section titled "Data Availability" at the end of the Methods section or main text (if no Methods). More information on this policy, is available at <http://www.nature.com/authors/policies/data/data-availability-statements-data-citations.pdf>.

If a community resource is unavailable, data can be submitted to generalist repositories such as <https://figshare.com/> or <http://datadryad.org/> Dryad Digital Repository. Please provide a unique identifier for the data (for example a DOI or a permanent URL) in the data availability statement, if possible. If the repository does not provide identifiers, we encourage authors to supply the search terms that will return the data. For data that have been obtained from publically available sources, please provide a URL and the specific data product name in the data availability statement. Data with a DOI should be further cited in the methods reference section.

REVIEWER COMMENTS:

Reviewer #1 (Remarks to the Author):

The study addresses an important question of public (host) attitudes towards climate migrants using a well-designed conjoint experiment to understand the mechanisms. The study fills an important gap of rural-to-rural migration in developing contexts. While a robust study that contributes to the study of migration attitudes (especially empirically), especially around climate in the Global South, there are concerns that should be addressed to improve the paper.

#Theory

Mechanism. The three mechanisms are presented as distinct but 'mutually enforcing' (p. 3). But there's a lack of clear theoretical prediction around when & how these mechanisms interact. Do they operate independently or does one mechanism (e.g., empathy) mediate another (e.g., deservingness)?

Empirics

Sampling bias. Recruiting respondents with migrant neighbours biases the sample selection as they've already been 'treated' with contact – exposed to migrants & have potentially already become tolerant. Further, migrants likely would not settle where there are high levels of pre-existing negative attitudes towards migrants in the first place. Moreover, the results show that contact theory doesn't pan out but the study does not explore why this may be the case – there are conditions of contact (e.g., frequency, quality) that could have enabled contact theory to hold true but given the measure is 'exposure', these nuances have not been captured.

Confounder. Are people who have experienced environmental hardship (erosion) less economically well-off? The 56% of respondents who reported losing home to erosion are more likely to be economic worse off than those who have not, so 'experiential empathy' would, instead of the mechanism of 'experience', be explained by economic similarity or class solidarity. Further, existing research shows lower social class exhibit more prosocial behaviour compared to high class individuals (e.g., Piff 2017), further adding to a confounding relationship.

High baseline levels. It seems that there's a high level of baseline acceptance of migrants which may indicate there are local cultural norms, religious frameworks, etc in rural Bangladesh that might not be universal beyond the region.

Direct measurement of the mechanisms. The study infers from the conjoint dimensions the mechanisms activated (deservingness, empathy, contact). Direct measurements could strengthen the causal claims. Relatedly, the question on why conjoint experiment is a better method to capture the public's migration attitudes than direct questions is currently missing and would strengthen the motivation for the paper.

Reviewer #2 (Remarks to the Author):

Review for: "People Like Us": Empathy and Acceptance of Climate Migrants in Rural Bangladesh
Manuscript ID: COMMSENV-25-2920-T

Thank you for giving me the opportunity to read and review this manuscript. This study is timely, interesting and well-presented both in theory and empirical analysis.

The study explores aspects of attitudes and emotional affect towards internally displaced environmental migrants. The findings reveal that moral judgments and experiential closeness can promote inclusive attitudes even in resource-limited contexts, carrying important implications for strengthening societal resilience to climate change. There are clear contributions to the existing literature in terms of internal mobility from urban-to-urban environments. The authors employ a conjoint survey experiment is that suits the research question and offer ample justification of their choice.

I think this paper could be revised and resubmitted. Here are some suggestions that the authors could consider incorporating in their study.

- The authors do a great job in the introduction motivating the paper and explaining environmental mobility. They also position clearly the importance of this study in the intersection of political, economic and social aspects. The authors show the contribution in relation to attitudes, but they also refer to emotions such as deservingness and empathy. The authors could also engage with some environmental psychology literature (next to social psychology) to strengthen the contribution but also the audience of this paper.

Some literature suggestions:

Briciu, B., 2020. "Anyone can become a refugee:" strategies for empathic concern in activist documentaries on migration. *Emotion, space and society*, 37, p.100737.

Wang, S., Leviston, Z., Hurlstone, M., Lawrence, C. and Walker, I., 2018. Emotions predict policy support: Why it matters how people feel about climate change. *Global Environmental Change*, 50, pp.25-40.

- There is also a sentence indicating the urge to look at rural societies and particularly mobility from a rural to another rural society and wonder whether the authors could slightly expand here to explain the importance of this focus in comparison to urban to urban or rural to urban. For example, it is likely easier for people prior living in a rural environment to move to another rural environment due to familiarity with the environment, cultural or employability reasons. I believe this is a major contribution of this study and ought to be highlighted.

- I suggest some light re-structuring to clearly show the existing scholarship and the theoretical framework of this study. That is, the authors could incorporate another section e.g., theoretical framework and expectations when presenting the mechanisms (currently in the introduction). The authors have already presented the existing literature there, but they could

also present/develop better their expectations justifying the choice of attributes. This would give the reader a section combining literature review and theory.

- The authors will have to be clearer about the number of participants, it seems like a 3x3 design to make it to 936 participants but not shown in the manuscript. Also, what are the 256 residents in northern Bangladesh? The B.2 section in the appendix could be more detailed if the authors would prefer to have this information in the supplementary material. The B.2 section only refers to the existence of a technical issue.
- Consider better explaining the disputes question. This is not part of the mechanisms, is this ultimately a control variable? Please clarify.
- I also suggest the authors to deepen the discussion of the findings whilst they put a lot of emphasis in describing all the findings. If there is lack of space, possibly focus on the core results but make sure there is a thorough discussion. For example, in the subgroups analysis, participants who have lost their house due to the erosion they have direct cognitive and emotion proximity to environmental migrants rather than economic migrants. This is interesting as the participants seem believe that environmental misfortune may be more important than generally economic misfortune. How and why do people justify/accept somebody who lost their house due to environmental hazard and not an economic hazard? Do possibly people assume that the economic hazard is personally driven where people had choices or made the wrong choices, but they do not have a choice with environmental hazards? This discussion of this result is also brief in the discussion section.
- I appreciate the generalizability section and effort the authors have put to verify their sample. A question raised here is if this study and the assumptions made could be applied to another country beyond Bangladesh and what do we learn from these, policy wise?

** Visit Nature Portfolio's author and referees' website at www.nature.com/authors for information about policies, services and author benefits**

Communications Earth & Environment is committed to improving transparency in authorship. As part of our efforts in this direction, we are now requesting that all authors identified as 'corresponding author' create and link their Open Researcher and Contributor Identifier (ORCID) with their account on the Manuscript Tracking System prior to acceptance. ORCID helps the scientific community achieve unambiguous attribution of all scholarly contributions. You can create and link your ORCID from the home page of the Manuscript Tracking System by clicking on 'Modify my Springer Nature account' and following the instructions in the link below. Please also inform all co-authors that they can add their ORCIDs to their accounts and that they must do so prior to acceptance.

Version 1:

Decision Letter:

Dear Dr Rudolph,

Please allow us to express our apologies for the delay in reaching a decision on your revised submission.

Your manuscript titled "'People Like Us': Empathy Informs Acceptance of Climate Migrants in Rural Bangladesh" has now been seen by our reviewers, whose comments appear below. In light of their advice we are delighted to say that we are happy, in principle, to publish a suitably revised version in Communications Earth & Environment.

We therefore invite you to revise your paper one last time to address the remaining concerns of our reviewers. At the same time we ask that you edit your manuscript to comply with our format requirements and to maximise the accessibility and therefore the impact of your work.

EDITORIAL REQUESTS:

Please address Reviewer 1's remaining concerns.

Please review our specific editorial comments and requests regarding your manuscript in the attached "Editorial Requests

Table".

****Please take care to match our formatting and policy requirements. We will check revised manuscript and return manuscripts that do not comply. Such requests will lead to delays. ****

SUBMISSION INFORMATION:

OPEN ACCESS:

Communications Earth & Environment is a fully open access journal. Articles are made freely accessible on publication. For further information about article processing charges, open access funding, and advice and support from Nature Portfolio, please visit <https://www.nature.com/commsenv/open-access>

Link Redacted

We hope to hear from you soon. Please note that our editorial offices will be closed or operate at reduced capacity until January 5, 2026.

Best regards,

Yann Benetreau, PhD
Consulting Editor, Communications Earth & Environment
Deputy Editor, Communications Sustainability
Nature Portfolio
ORCID: 0000-0002-1897-0887
New York Office

REVIEWERS' COMMENTS:

Reviewer #1 (Remarks to the Author):

Many thanks to the authors for a thorough and thoughtful responses to the initial comments. I appreciate the substantial effort the authors have put into revising the manuscript, and I acknowledge that many of my concerns have been addressed. The paper is considerably stronger in its theoretical framing, empirical robustness checks, and transparency about limitations. I have two final comments.

First, the conceptualisation and empirical evaluation of the contact mechanism remains unconvincing. The core measure ('new families arriving in the village') is a coarse proxy for interpersonal contact and even the added subgroup ('close migrant friends') does not isolate contact with environmental migrants. Additional data collection is likely not feasible at this stage; however, the paper could reposition this mechanism as a tentative part of the framework.

Second, the authors use multiple testing without correction. The manuscript tests 20+ hypotheses across multiple subgroups without any correction for multiple comparisons. This is particularly problematic as several of the key findings rest on marginal p-values that would not survive standard corrections – e.g., H3 (deservingness moderates religious penalty): $p=.146$, H4 experiential proximity (the key empathy finding) is at $p=.122$ (line 354). This is important especially since the title claims 'empathy informs acceptance' but rests on $p=.122$ evidence. The paper need not undertake extensive statistical corrections, but the authors should apply multiple testing corrections and report which findings remain robust and/or consider moderating claims as exploratory than confirmatory.

With these remaining clarifications and adjustments, the manuscript would be well positioned for publication.

Reviewer #2 (Remarks to the Author):

I am very satisfied with the way the authors have addressed all the suggested points. It is evident that they have invested considerable effort in developing this research, and the outcome is commendable. The study's contribution is clear, and I am confident that both academic scholarship and the policy sector will benefit significantly from its findings.

** Visit Nature Portfolio's author and reviewers' website at www.nature.com/authors for information about policies, services and author benefits**

Response Memo

‘People Like Us’: Empathy Informs Acceptance of Climate Migrants in Rural Bangladesh

November 3, 2025

Dear Professor Dasandi, dear Dr Benetreau, dear Referees,

we thank you for the work entailed in assessing our manuscript “‘People Like Us’: Empathy and Acceptance of Climate Migrants in Rural Bangladesh” (COMMSENV-25-2920-T). We are grateful for your very helpful feedback on the previous version of the manuscript. In light of your suggestions, we have carefully and extensively revised the paper and remain committed to implementing any additional changes needed to make it acceptable for publication in *Communications Earth & Environment*.

In what follows, we reproduce the comments in order (in standard font, with cross-numbering) and respond to them directly (in **bold** font) noting the corresponding changes made to the manuscript. For ease of refereeing, we incorporate core revisions of the manuscript directly in this memo (in *italics*). In the revised manuscript file, we highlighted paragraphs with substantive changes in **red** font. As requested by the editor, we also provide a tracked-changes version in a separate file.¹

Our replies to the Editor begin on page 2, to Reviewer #1 on page 4, and to Reviewer #2 on page 14.

We very much look forward to hearing from you again.

Yours sincerely,

The authors

¹Please note that the track-change document depicts changes to the manuscript text only (changes to references and links within the document did not compile, unfortunately).

Comments by Editor

E-1 (...) We therefore invite you to revise and resubmit your manuscript, along with a point-by-point response that takes into account the points raised. Please highlight all changes in the manuscript text file.

Reply: We thank the editors for the opportunity to revise and resubmit our work.

E-2 In particular, we wanted to flag Reviewer 1’s final comment on direct measurements helping to strengthen support for causal claims. This is an important point, and one that we feel is particularly important that you address.

Reply: We thank the editors for emphasizing this point (see also our reply to comment R1-6). In Methods (section 6.3), we clarify how the conjoint experimental design strengthens inference about the mechanisms in our framework. In our setting, deservingness plausibly underlines the higher favorability of environmental migrants. Although we did not field direct ‘reasons’ questions (which can invite post-hoc rationalization), the randomized comparison of (conjoint) profiles varying the reason for migration alongside attributes proxying economic competition (occupation) and cultural threat (religion, origin), allows to interpret the 21%-point higher choice probability for environmental vs. economic migrants as a preference structure net of these alternative channels. Similar conjoint studies draw the same deservingness interpretation (e.g. Arias and Blair, 2022; Spilker et al., 2020; Bansak, Hainmueller, and Hangartner, 2016). Regarding experiential proximity and contact, we rely on direct and self-reported measures and find that responses differ for experience, but not contact.

E-3 In order for the study to be published in Communications Earth & Environment, your study should provide compelling novel insights on the acceptance of climate migrants in rural Bangladesh and the distinct role of each mechanistic pathway (deservingness, empathy, contact). (...)

Reply: We thank the editors for flagging this point. Please also see our reply to R2-3. In short, we have expanded the introduction to emphasize the importance of rural–rural mobility and to reinforce the positioning of our study. We also revised the concluding paragraph of introduction to clarify contributions, and the distinct roles of the three mechanisms (deservingness, empathy, contact). In addition, we clarified the hypotheses tied to each mechanism and sharpened the Results/Discussion to highlight how each mechanism relates to acceptance in rural Bangladesh.

E-4 (...) An updated and completed version of our Reporting Summary must be uploaded with the revised manuscript. (...) Furthermore, please align your manuscript with our format requirements (...).

Reply: We thank the editor for guiding us to the reporting summaries and checklists. We uploaded the reporting summaries along with our resubmission, and revised the manuscript subsequent to the format requirements. In detail, we

- changed the title to communicate our main finding
- shortened the abstract
- revised the introduction to contain a concluding summary paragraph
- separated Main and Methods
- renamed the Appendix as Supplementary Information and ensured all SI material is referenced
- incorporated footnotes into the manuscript or Supplementary Information
- added concluding statements

E-5 (...) DATA: Communications Earth & Environment endorses the principles of the Enabling FAIR data project. We ask authors to make the data that support their conclusions available in permanent, publicly accessible data repositories (, ... and) include a section titled "Data Availability" at the end of the Methods section or main text.

Reply: We thank the editors for raising this point. We are committed to transparent and reproducible research and will provide full replication files. We have already assembled these, and propose to deposit them once reviewers have no additional data-related requests, on Harvard Dataverse under <https://doi.org/10.7910/DVN/MDC2EG> (doi reserved, not live yet). Corresponding code and data availability statements are added to the manuscript.

Comments by Reviewer #1

R1-1 The study addresses an important question of public (host) attitudes towards climate migrants using a well-designed conjoint experiment to understand the mechanisms. The study fills an important gap of rural-to-rural migration in developing contexts. While a robust study that contributes to the study of migration attitudes (especially empirically), especially around climate in the Global South, there are concerns that should be addressed to improve the paper.

Reply: We thank R1 for the thorough assessment of our manuscript. Below, we provide detailed replies to the very helpful comments provided by R1.

R1-2 #Theory Mechanism. The three mechanisms are presented as distinct but ‘mutually enforcing’ (p. 3). But there’s a lack of clear theoretical prediction around when & how these mechanisms interact. Do they operate independently or does one mechanism (e.g., empathy) mediate another (e.g., deservingness)?

Reply: We appreciate this helpful comment by R1. In the revision, we made three clarifications: We first removed the “mutually reinforcing” phrasing and now treat the three mechanisms as analytically distinct considerations, namely deservingness (involuntariness), empathy (social and spatial similarity; experiential proximity), and interpersonal contact. This is stated at the outset of the theory section:

Each mechanism captures a distinct form of psychological and social proximity that influences how host populations evaluate newcomers. Rather than assuming that these mechanisms are mutually reinforcing, we treat them as analytically distinct and state the conditions under which they interact. (LL. 67ff.)

As before, we first introduce deservingness (now H1) as main effect of environmental reason of migration, and then lay out independent main effects for religion, distance, and occupation (H2). Next, also the more complex expectations stand on their own in the framework.

First, we add a short bridge that provides a single, cross-cutting moderation prediction (H3) focusing on empathy. The logic ties directly to the deservingness heuristic and explains why it should temper penalties on otherwise disfavored attributes:

Beyond these independent effects, deservingness should do more than lift average support for involuntary migrants. In particular, by lowering blame, widening moral concern, and increasing the willingness to extend help across group boundaries, it should soften exclusionary responses to otherwise disfavored attributes (Spilker et al., 2020; Bansak, Hainmueller, and Hangartner, 2016; Weiner, 1995; Verkuyten, Mepham, and Kros, 2018). Consequently, even when migrants differ in religion, originate from more distant areas,

or hold low-status occupations, they may be judged less harshly if their displacement is clearly beyond their control. We therefore expect the environmental-motive (deservingness) cue to operate as a moderating force, attenuating the penalties associated with social and economic dissimilarity (H3). (LL. 124ff.)

Next, empathy should also derive from experiential proximity (erosion loss, erosion-driven moves, migration histories), and we clarify that we expect a premium for environment as reason for migration from this:

In addition, and because the salience of involuntariness most likely differs across respondents, the assessment of environmental migrants should vary with hosts' own exposure to environmental risk and mobility. [...] By making need salient and inviting perspective taking, experiential proximity strengthens support for involuntary displaced migrants; we therefore expect the deservingness effect to be strongest among respondents with greater experiential proximity. (LL. 133ff)

This positions experiential proximity as a heterogeneity condition on H1.

Finally, interpersonal contact is retained as a conceptually distinct mechanism by which we expect heterogeneity in the environmental-motive effect (H5):

The third, conceptually distinct mechanism is interpersonal contact with migration. [...] Conceptually, the contact mechanism complements deservingness and social, spatial, and experiential proximity by highlighting that judgments reflect both perceptions of migrants and hosts' lived experiences. In the environmental case, contact can make the involuntariness of displacement more salient and credible, for instance, through observable loss and first-hand narratives, which may strengthen or, under scarcity, weaken the weight given to the environmental-motive signal (H5). (LL. 156ff.)

This clarification yields five concrete predictions that align one-to-one with the empirical section:

H1: Main effect of deservingness (environmental > economic/social).

H2: Independent main effects of religion (in-group), distance (near > far), and occupation (higher-status > lower-status).

H3: Interaction effects: deservingness (involuntariness) attenuates penalties from out-group religion, distant origin, and low occupational status.

H4/5: Heterogeneity in subgroups: experiential proximity (H4) and interpersonal contact (H5) condition the environmental-motive effect.

R1-3 # Empirics Sampling bias. Recruiting respondents with migrant neighbours biases the sample selection as they've already been 'treated' with contact – exposed to migrants & have potentially already become tolerant. Further, migrants likely would not settle where there are high levels of pre-existing negative attitudes towards migrants in the first place. Moreover, the results show that contact theory doesn't pan out but the study does not explore why this may be the case – there are conditions of contact (e.g., frequency, quality)

that could have enabled contact theory to hold true but given the measure is ‘exposure’, these nuances have not been captured.

Reply: We thank R1 for raising this important point. First, on how contact theory in our case, we do not expect general contact to uniformly increase acceptance. Rather, following the deservingness logic, we expect contact to matter specifically in relation to the environmental-motive cue. When displacement is attributed to environmental loss, interpersonal interaction can render involuntariness more vivid and credible, for instance, through observable erosion damage or first-hand narratives, thereby heightening empathic concern and willingness to extend help. By contrast, in resource-constrained settings, such contact can also underscore the material costs of accommodating newcomers, thereby dampening support. In this sense, contact operates as a moderator of the environmental-motive effect, not as an independent predictor. This explains why we expect its role to be specific to environmental migrants and why we test it as H4. (LL.170ff.)

Second, to address the concern about potential sample bias, we conducted additional analyses. Specifically, we use on a survey item asking whether respondents have migrants as close friends (vs. not), which allows us to test whether the quality of contact influences our results. We report the corresponding marginal means for these two subgroups in Figure RM.1, now also included in the Supplementary Material as Figure A7. Extending our analysis in this way, we find that both subgroups show similarly higher favorability for environmental compared to economic migrants. Hence, neither the extensive margin (perceived arrival of new families: yes/no), nor the intensive margin (migrants as close friends: yes/no) yield meaningful differences on the migrant-type attribute. While informative, we now provide for a more cautious interpretation of these results in the manuscript. First, as noted by R1, sampling bias might affect the real-world variation we observe, and immigration into communities where both objectively and subjectively no new families arrived in the last time might yield different results. Second, we lack information on the quality of contact with particular types of migrants, especially whether respondents have environmental migrants as ‘close friends’, which would more directly activate our proposed mechanism. Lastly, respondents with and without migrants as close friends may differ on other dimensions, such as education, social status, or their own mobility. Nonetheless, we believe that including Figure RM.1 in the Supplementary Material strengthens our manuscript by further supporting the robustness of our findings, and we are grateful to R1 for directing us to this improvement.

Figure RM.1. Marginal means for subgroups of respondents that have migrants as close friends ($N = 210$) vs. not as close friends ($N = 156$). Error bars indicate 95% confidence intervals from respondent-clustered standard errors. The exact survey question read: “About how many of your close friends in [village] came as migrants from other places to [village] in the last 5 years?” Respondents who answered that none of their friends came as migrants were counted as having no migrants as close friends, while respondents who answered that a few, about half, many, or all of their friends have come as migrants were counted as having migrants as close friends. As this question was part of an additional priming experiment (see Methods section 6.4), it was posed only to a random subset of the initial study sample.

Next to new SM Figure A.7, we now added to the Discussion section of the manuscript:

Note, however, that this finding could also reflect our sampling strategy, which specifically targets migrant-receiving areas. (see Lines 407f.); and

However, when we extend the analysis to compare respondents who report having migrants as close friends (vs. not), we likewise observe no differences in favorability by reason for migration (while a slightly lower religious bias emerges, see Supplementary Figure A.7). Future research could investigate whether close contact, specifically with environmental migrants, meaningfully differentiates preferences. (see Lines 501ff.).

R1-4 Confounder. Are people who have experienced environmental hardship (erosion) less economically well-off? The 56% of respondents who reported losing home to erosion are more likely to be economic worse off than those who have not, so ‘experiential empathy’ would, instead of the mechanism of ‘experience’, be explained by economic similarity or class solidarity. Further, existing research shows lower social class exhibit more prosocial behaviour compared to high class individuals (e.g., Piff 2017), further adding to a confounding relationship.

Reply: We thank R1 for this helpful comment. We agree that confounders could contribute to the patterns we observe, although we deem it unlikely that they drive our results. When we compared respondents who reported past house loss with others on (self-reported) objective and subjective income and on education, we found only small correlations (see Figure RM.2 below, which corresponds to SM Figure A.6 in the manuscript). Nonetheless, because our research design is not tailored to causally differentiating respondents by experience of this moderator, we now interpret these findings more cautiously. We added the following to the manuscript:

Note, however, that potential confounders should be assessed in future research. For example, experiencing environmental hardship may correlate with social class – and lower social class in itself appears to be correlated with prosocial behavior (Piff and Robinson, 2017). While our design is not tailored to causally differentiate respondents by the moderator ‘environmental hardship’, we consider major confounding unlikely. First, comparisons on observable markers of economic well-being, such as (objective and subjective) household income or education, seem largely comparable in our sample between individuals reporting environmental hardship and those who do not (see Supplementary Figure A.6). Second, riverbank erosion affects longstanding communities in a quasi-random fashion, making economic well-being an unlikely strong predictor of affectedness (Rudolph, Koubi, and Freihardt, 2025).

Figure RM.2. Relative frequencies of indicators of economic well-being and social class by house loss due to erosion for the conjoint sample used in Figure 4 Panel (b) ($N_{respondents} = 155$). (a) Household income measured on a 10-point scale. (b) Self-reported ability to sustain household on current income measured on a 5-point scale. (c) Education measured on three levels. NAs are reported for the sake of transparency. We found no significant correlation between house loss due to erosion and household income ($r = -0.047$, $p = 0.585$), a statistically significant but weak correlation between house loss and respondents' self-reported ability to sustain their household on current income ($r = 0.177$, $p = 0.028$) and no significant correlation between education and house loss ($r = -0.035$, $p = 0.667$). Correlations are estimated between a binary variable measuring house loss and the numerical values of the respective variables indicated on the x-axes (NAs excluded).

R1-5 High baseline levels. It seems that there's a high level of baseline acceptance of migrants which may indicate there are local cultural norms, religious frameworks, etc in rural Bangladesh that might not be universal beyond the region.

Reply: We thank R1 for this point, which is indeed intriguing – and which we now address in our generalizability section (section 3.4). In cross-national comparison, the Bangladeshi population also appears to exhibit above aver-

age acceptance of migrants (Gu, Zhang, and Lin, 2022, Fig. 3). Even with high baseline acceptance, however, our survey provides sufficient variation in acceptance. We therefore added a subgroup analysis comparing respondents who state that they would accept migrants as neighbors vs. those that do not state so. Results are shown in Figure RM.3 and are added to the Supplementary Information as Supplementary Figure A.5. They do not indicate that our results depend on baseline acceptance levels. Accordingly, we added the following discussion to section 3.4:

Of course, this does not imply that our findings necessarily generalize to other rural contexts in the Global South. Our study population is distinctive in combining high spatial and experiential proximity with generally low levels of migrant hostility (see Section 3.1). Such comparatively low migrant hostility also appears in Bangladeshi samples in cross-country data (see, e.g., Gu, Zhang, and Lin, 2022, Fig. 3). Hence, our context may be specific along multiple dimensions. This context specificity is, however, consistent with the broader literature, which reports that the perceived deservingness of climate-displaced migrants appears in Global North settings (see for evidence from the US and Germany Arias and Blair, 2022; Helbling, 2020), but appears weaker in some urban Global South environments (Spilker et al., 2020). Notably, these country cases display similar average acceptance levels (see Gu, Zhang, and Lin, 2022, Fig. 3), underscoring the role of additional moderating factors, such as spatial and experiential proximity, that we emphasize here. Future research could use our design as a template to further probe this heterogeneity across settings. In addition, in our case, differentiating respondents by their acceptance of migrants does not substantially change favorability toward reasons for migration (see Supplementary Figure A.5). (LL.383ff.)

And in the Conclusion, we added:

While our evidence comes from rural Bangladesh, similar qualitative patterns are most likely where rural receiving communities face visible, involuntary environmental loss and relocation occurs nearby through kinship and shared livelihoods. In such settings, a deservingness premium should emerge and be amplified by proximity; where baseline acceptance is lower or cultural/spatial distance is greater, for example, urban or long-distance moves, magnitudes may attenuate or differ. For policy, this implies facilitating near-site relocation and family reunification, communicating the involuntary/need-based nature of displacement, and pairing newcomer support with host-community investments in services and livelihoods to ease resource competition. (LL.551ff.)

Figure RM.3. Marginal means for subgroups of respondents that said that they would accept migrants as neighbors ($N = 450$) vs. not ($N = 114$). Error bars indicate 95% confidence intervals from respondent-clustered standard errors. Acceptance of migrants is measured by binarizing replies to the question “Please tell me whether you would like having people from outside (village name), meaning not from this area, as neighbors (...)” (see Supplementary Table A.4 for additional details). Respondents that said that they would strongly like or somewhat like migrants as their neighbors were counted as accepting migrants while respondents saying that they would not care, somewhat dislike or strongly dislike migrants as neighbors were categorized as not accepting migrants.

R1-6 Direct measurement of the mechanisms. The study infers from the conjoint dimensions the mechanisms activated (deservingness, empathy, contact). Direct measurements could strengthen the causal claims.

Reply: We thank R1 for this intriguing question. We read R1’s query as asking whether the patterns in migrant preferences we observe could be observationally equivalent to alternative mechanisms – and whether direct survey questions about *why* migrant acceptance is higher for environmental, as compared to economic/family reasons, would have strengthened our claims. We agree that such questions would have been a valuable addition to our questionnaire, which we unfortunately did not think of at the time. Hence, direct survey items on reasons for higher acceptance are not available to us.

However, we still believe that the conjoint experimental evidence we provide is strongly points to the proposed mechanisms. This is for four reasons:

First, regarding deservingness, the comparison of conjoint profiles that vary the reason for migration implicitly controls for two core alternative channels of migrant acceptance: economic competition (proxied by occupation) and cultural threat (proxied by religion and origin). We can therefore interpret the 21%-point higher choice probability for environmental relative to economic migrants as a preference structure net of these alternative channels. The underlying problem addressed here is so-called ‘masking’ (Bansak, Hainmueller, Hopkins, et al., 2021), whereby respondents infer multiple characteristics from a single attribute (i.e., not only deservingness). We believe that our design, which simultaneously fields these attributes, follows core explanations in the literature (economic; cultural/symbolic, see Hainmueller and Hopkins (2014)), and similar conjoint studies have drawn comparable deservingness interpretations (e.g. Arias and Blair, 2022; Bansak, Hainmueller, and Hangartner, 2016). We concede, however, that favorability towards environmental relative to economic migrants could also relate to attributes we did not include.

(The explanation above is added to the Methods section (see section 6.3).

Second, direct inquiry into motivations has its own problems (which is why we opted to infer them indirectly from the conjoint experiment). Most notably, humans tend to engage in post-hoc rationalizing (Summers, 2017; Cushman, 2020), complicating inference from self-reported motivations (Citrin et al., 1997). Hence, indicating that a migrant is “undeserving” might go hand-in-hand with perceiving the migrant as an economic threat, and vice versa.

Third, direct questions are more prone to social desirability. By contrast, comparing conjoint profiles that vary the reason of migration provides a direct measure of favorability towards environmental reasons, averaging over the distribution of the remaining attributes.

Fourth, for experiential proximity and contact, we rely on self-reported measures and find that conjoint responses differ in case of the contact, but not experience. However, this inference is prone to omitted variable bias – as contact/experience may correlate with third factors that actually drive the observed preference differences. We discuss this more extensively in our reply to comment R1-4.

R1-7 Relatedly, the question on why conjoint experiment is a better method to capture the public’s migration attitudes than direct questions is currently missing and would strengthen the motivation for the paper.

Reply: We thank R1 for this remark and have updated our the section on research design accordingly. Survey experiments have become an increasingly prominent part of migration studies since the experimental turn of the 2010s (Allen and Vargas-Silva, 2024). One key reason is social desirability. Direct questions on migrant acceptance are prone to this bias and tend to underestimate hostility against migrants (Rinken et al., 2021). Conjoint survey experiments help to mitigate this bias by concealing researcher objectives (Mummolo and Peterson, 2019) and respondents’ preferences on single attributes (Horiuchi, Markovich, and Yamamoto, 2022; Dahl, 2018), are relevantly mitigating this bias, though. This is particularly true for the binary-profile forced-choice design we employ, which has been shown to generalize well to population preferences on migrant acceptance (Hainmueller, Hangartner, and Yamamoto, 2015). Of course, because conjoint profiles are hypothetical, scenarios must be relatable for respondents (Bladel et al., 2008). This condition is met in our sample, and recent evidence indicates that hypotheticality does not systematically affect responses in conjoint experiments (Brutger et al., 2023).

This is now implemented in the manuscript as (see Methods section 6.3):

In addition, conjoint experiments have been shown to relevantly mitigate social desirability bias, which direct questions on migrant acceptance are likely prone to (Rinken et al., 2021), by concealing both researcher objectives (Mummolo and Peterson, 2019) and respondents’ preferences on singular attributes (Horiuchi, Markovich, and Yamamoto, 2022; Dahl, 2018). Moreover, the binary-profile forced-choice design we employ has been shown to generalize well to actual population preferences on migrant acceptance (Hainmueller, Hangartner, and Yamamoto, 2015). (LL.633ff.)

Overall, we once again cordially thank R1. The revisions suggested by R1 have helped us to considerably strengthen the contributions of our manuscript.

Comments by Reviewer #2

R2-1 Thank you for giving me the opportunity to read and review this manuscript. This study is timely, interesting and well- presented both in theory and empirical analysis. The study explores aspects of attitudes and emotional affect towards internally displaced environmental migrants. The findings reveal that moral judgments and experiential closeness can promote inclusive attitudes even in resource-limited contexts, carrying important implications for strengthening societal resilience to climate change. There are clear contributions to the existing literature in terms of internal mobility from urban-to-urban environments. The authors employ a conjoint survey experiment is that suits the research question and offer ample justification of their choice.

I think this paper could be revised and resubmitted. Here are some suggestions that the authors could consider incorporating in their study.

Reply: We thank R2 for the detailed assessment of our manuscript. We provide below a point-by-point response to the very constructive comments raised by R2.

R2-2 The authors do a great job in the introduction motivating the paper and explaining environmental mobility. They also position clearly the importance of this study in the intersection of political, economic and social aspects. The authors show the contribution in relation to attitudes, but they also refer to emotions such as deservingness and empathy. The authors could also engage with some environmental psychology literature (next to social psychology) to strengthen the contribution but also the audience of this paper. Some literature suggestions: Briciu, B., 2020. "Anyone can become a refugee:" strategies for empathic concern in activist documentaries on migration. *Emotion, space and society*, 37, p.100737. Wang, S., Leviston, Z., Hurlstone, M., Lawrence, C. and Walker, I., 2018. Emotions predict policy support: Why it matters how people feel about climate change. *Global Environmental Change*, 50, pp.25-40.

Reply: We appreciate this suggestion. We have incorporated the recommended environmental psychology literature directly into the theory section as follows:

From an environmental-psychology perspective, affective reactions, especially climate worry or concern, predict support for climate policies (Wang et al., 2018). Empathic concern is especially likely when loss and vulnerability are rendered vivid through shared hazard exposure or narratives that personalize displacement (Briciu, 2020). (LL. 149ff.)

These additions strengthen the conceptual framework by connecting our empathy and deservingness mechanisms more explicitly to affective processes highlighted in environmental psychology.

R2-3 There is also a sentence indicating the urge to look at rural societies and particularly mobility from a rural to another rural society and wonder whether the authors could

slightly expand here to explain the importance of this focus in comparison to urban to urban or rural to urban. For example, it is likely easier for people prior living in a rural environment to move to another rural environment due to familiarity with the environment, cultural or employability reasons. I believe this is a major contribution of this study and ought be highlighted.

Reply: We appreciate this helpful suggestion. In response, we have expanded the introduction to emphasize the importance of rural–rural mobility and to reinforce the positioning of our study, adding the following texts: In the introduction:

This focus is especially salient because environmental migration is predominantly internal and often short-distance, frequently rural-to-rural (Rigaud et al., 2018; Cattaneo et al., 2019; Hoffmann et al., 2020). Micro-level studies corroborate this pattern specifically for Bangladesh, where actual environmentally induced moves are primarily rural-to-rural (Freihardt, 2025; Paul et al., 2022), and households facing erosion stress disproportionately aspire to relocate to nearby rural localities rather than urban centers (Rudolph, Koubi, and Freihardt, 2025; Paul et al., 2022). These findings suggest that environmental mobility does not conform to a simple rural–urban transition; rather, it reflects strategies to minimize disruption by remaining in familiar settings, including sustaining agrarian livelihoods and norms, leveraging transferable skills for local work, and relying on kinship and social networks that structure settlement, thereby reducing cultural distance. Against this backdrop, examining attitudes in rural receiving areas is crucial, as these communities are likely to remain primary destinations for environmental migrants in Bangladesh and many similarly affected contexts in the Global South for internal, short-distance moves. (LL. 49ff.)

In the Introduction- Contributions:

We emphasize rural destinations because prior rural residence fosters familiarity with local socio-ecological settings, reducing cultural distance despite resource constraints. Testing for a deservingness premium and its amplification by proximity in this context would provide strong evidence for the proposed mechanisms. (LL. 190ff.)

We believe that these additions strengthen the positioning of our study and underscore its broader relevance.

R2-4 I suggest some light re-structuring to clearly show the existing scholarship and the theoretical framework of this study. That is, the authors could incorporate another section e.g., theoretical framework and expectations when presenting the mechanisms (currently in the introduction). The authors have already presented the existing literature there, but they could also present/develop better their expectations justifying the choice of attributes. This would give the reader a section combining literature review and theory.

Reply: Thank you for this helpful suggestion. We considered creating a standalone theory section, but because the directly relevant literature is very lim-

ited, a separate section would largely duplicate material from the Introduction without improving clarity. To maintain a clear line of argumentation from the research gap to our mechanisms and expectations and to save space, we retain the theoretical discussion within the Introduction. However, to address your request, we made the expectations explicit in one sentence at the end of the discussion of each mechanism. We believe that this preserves brevity while making our framework and its empirical implications fully transparent.

R2-5 The authors will have to be clearer about the number of participants, it seems like a 3x3 design to make it to 936 participants but not shown in the manuscript. Also, what are the 256 residents in northern Bangladesh?

Reply: We thank R2 for pointing this out and allowing us to clarify. To briefly explain our study N, we conducted the study with 256 residents in northern Bangladesh. Of these, 165 participated in the conjoint survey experiment – the shortfall from 256 reflects technical issues with displaying the conjoint attribute pictures in the Qualtrics Offline Survey App during the interviews, which we logged in our preregistration. Of the 165, 156 respondents understood and completed the conjoint task. Each participant completed three conjoint rounds, yielding $2 \text{ profiles} \times 3 \text{ tasks} \times 156 \text{ respondents} = 936$ profile evaluations. For avoidance of doubt, this is not a 3×3 participant design; the “3” refers to the number of conjoint tasks per respondent. We use all 256 respondents for the analyses of general attitudinal questions (see section 3.1, descriptives), with results presented in Supplementary Tables A.2 - A.4. We use the 156 respondents who completed the conjoint task (=936 observations) for all conjoint experimental analyses. This treats the missingness as missing-at-random, which is plausible given that technical errors, rather than respondent characteristics, prevented data collection. Supplementary Table A.1 provides evidence consistent with this assumption. It compares summary statistics for the full sample and the conjoint subsample and indicates high comparability. Please also note that the generalizability section (section 3.4) benchmarks the 156 conjoint respondents to a representative sample of the riverbank population and indicates no concern with selection. We have reworked how we present these numbers in the abstract, case section and Methods section following this comment (and R1), to prevent any confusion between participants and observations.

R2-6 The B.2 section in the appendix could be more detailed if the authors would prefer to have this information in the supplementary material. The B.2 section only refers to the existence of a technical issue.

Reply: We thank R2 for this comment. Appendix B.2 reproduced, verbatim, our OSF pre-registration – the copy-pasted OSF Project Summary (old

manuscript lines 785-794) and the PDF uploaded to OSF (old manuscript appendix page 12 ff.) – to ensure full transparency. To avoid redundancies and potential inconsistencies across versions, we have deleted Appendix B.2 and replaced it with an anonymized reference to the pre-registration.

R2-7 Consider better explaining the disputes question. This is not part of the mechanisms, is this ultimately a control variable? Please clarify.

Reply: We thank R2 for raising this point. We have streamlined section “3.1 Descriptives” accordingly. First, we report descriptives on migrant attitudes (Appendix Table A2), indicating a generally sympathetic stance towards migrants and low levels of grievances among natives. Second, this helps explain the high levels of general acceptance of migrants (see also comment R1-5). Taken together, we treat the dispute question as contextual description of the receiving environment, that is, an indicator of whether the local climate appears hostile or calm toward newcomers, and we establish tentative evidence for an environment seemingly not hostile to migrants in general. It is not part of our theoretical mechanisms (deservingness, proximity, contact) and is not used as a control in the conjoint analyses. We hope the revised paragraphs make this clearer.

R2-8 I also suggest the authors to deepen the discussion of the findings whilst they put a lot of emphasis in describing all the findings. If there is lack of space, possibly focus on the core results but make sure there is a thorough discussion. For example, in the subgroups analysis, participants who have lost their house due to the erosion they have direct cognitive and emotion proximity to environmental migrants rather than economic migrants. This is interesting as the participants seem believe that environmental misfortune may be more important than generally economic misfortune. How and why do people justify/accept somebody who lost their house due to environmental hazard and not an economic hazard? Do possibly people assume that the economic hazard is personally driven where people had choices or made the wrong choices, but they do not have a choice with environmental hazards? This discussion of this result is also brief in the discussion section.

Reply: We thank R2 for this remark. We have revised the discussion section to deepen the contextualization of our findings in the existing literature. We also thank R2 for the good suggestion on possible explanations for the increased migrant support among respondents who have suffered from erosion themselves. We have now taken this up in the discussion section:

Hosts who have suffered from erosion themselves may recognize that environmental shocks lie largely outside affected households control. This, in turn, may contribute to more favorable assessments relative to economic migrants, whose hardship may be perceived as stemming from their own choices.

Further, we added a note on potential confounders (also in response to comment R1-4):

Note, however, that potential confounders should be assessed in future research. For example, experiencing environmental hardship may correlate with social class – and lower social class in itself appears to be correlated with prosocial behavior (Piff and Robinson, 2017). While our design is not tailored to causally differentiate respondents by the moderator ‘environmental hardship’, we consider major confounding unlikely. First, comparisons on observable markers of economic well-being, such as (objective and subjective) household income or education, seem largely comparable in our sample between individuals reporting environmental hardship and those who do not (see Supplementary Figure A.7). Second, riverbank erosion affects longstanding communities in a quasi-random fashion, making economic well-being an unlikely strong predictor of affectedness (Rudolph, Koubi, and Freihardt, 2025).

R2-9 I appreciate the generalizability section and effort the authors have put to verify their sample. A question raised here is if this study and the assumptions made could be applied to another country beyond Bangladesh and what do we learn from these, policy wise?

Reply: We thank R2 for raising the generalizability of our findings beyond Bangladesh; we see this as closely related to comment R1-5 on the high baseline migrant acceptance among our respondents. We address both points in the generalizability section (section 3.4) and now add the following discussion:

Of course, this does not imply that our findings necessarily generalize to other rural contexts in the Global South. Our study population is distinctive in combining high spatial and experiential proximity with generally low levels of migrant hostility (see Section 3.1). Such comparatively low migrant hostility also appears in Bangladeshi samples in cross-country data (see, e.g., Gu, Zhang, and Lin, 2022, Fig. 3). Hence, our context may be specific along multiple dimensions. This context specificity is, however, consistent with the broader literature, which reports that the perceived deservingness of climate-displaced migrants appears in Global North settings (see for evidence from the US and Germany Arias and Blair, 2022; Helbling, 2020), but appears weaker in some urban Global South environments (Spilker et al., 2020). Notably, these country cases display similar average acceptance levels (see Gu, Zhang, and Lin, 2022, Fig. 3), underscoring the role of additional moderating factors, such as spatial and experiential proximity, that we emphasize here. Future research could use our design as a template to further probe this heterogeneity across settings. (LL. 383ff.)

And in the Conclusion we state:

While our evidence comes from rural Bangladesh, similar qualitative patterns are most likely where rural receiving communities face visible, involuntary environmental loss and relocation occurs nearby through kinship and shared livelihoods. In such settings, a deservingness premium should emerge and be amplified by proximity; where baseline accep-

tance is lower or cultural/spatial distance is greater, for example, urban or long-distance moves, magnitudes may attenuate or differ. For policy, this implies facilitating near-site relocation and family reunification, communicating the involuntary/need-based nature of displacement, and pairing newcomer support with host-community investments in services and livelihoods to ease resource competition. (LL.551ff.)

Overall, we once again thank R2 very much. The revisions suggested by R2 have substantially strengthened the contributions of our manuscript.

Response Memo

Empathy Informs Host Community Attitudes toward Climate Migrants in Rural Bangladesh

January 19, 2026

Dear Dr Benetreau, dear Referees,

we thank you for the work entailed in assessing our manuscript “‘People Like Us’: Empathy and Acceptance of Climate Migrants in Rural Bangladesh” (COMMSENV-25-2920A) and are thrilled you were able to conditionally accept our manuscript. We have carefully revised the manuscript to incorporate your helpful editorial checklist, as well as the remaining suggestions by reviewer 1. Reviewer 2 was already fully satisfied. Of course, we remain committed to implementing any additional changes needed to make our manuscript acceptable for publication in *Communications Earth & Environment*.

In what follows, we reproduce the comments in order (in standard font, with cross-numbering) and respond to them directly (in **bold** font), noting the corresponding changes made to the manuscript. For ease of refereeing, we incorporate core revisions of the manuscript directly in this memo (in *italics*). As requested by the editor, we also provide a supplementary manuscript file with the filled editorial checklist.

Our replies to the Editor begin on page 2, to Reviewer #1 on page 3.

We very much look forward to hearing from you again.

Yours sincerely,

The authors

Comments by Editor

E-1 ... Please address Reviewer 1's remaining concerns.

Please see below for our detailed replies to R1's remaining, and as always, valuable comments. In short, we repositioned the contact mechanism as tentative in the framework; and we added multiple hypothesis corrections (interpretation subsequent false discovery rate thresholds following the Benjamini-Hochberg-procedure) as a new Methods section (last section of Methods).

E-2 ... Please review our specific editorial comments and requests regarding your manuscript in the attached "Editorial Requests Table". ...

Please find our detailed replies in the checklist we returned with filled right-hand column.

Overall, we want to cordially thank the editors for a careful, swift and supportive review process.

Comments by Reviewer #1

R1-1 Many thanks to the authors for a thorough and thoughtful responses to the initial comments. I appreciate the substantial effort the authors have put into revising the manuscript, and I acknowledge that many of my concerns have been addressed. The paper is considerably stronger in its theoretical framing, empirical robustness checks, and transparency about limitations. I have two final comments.

We thank R1 for the thoughtful consideration of our manuscript throughout the review process.

R1-2 First, the conceptualisation and empirical evaluation of the contact mechanism remains unconvincing. The core measure ('new families arriving in the village') is a coarse proxy for interpersonal contact and even the added subgroup ('close migrant friends') does not isolate contact with environmental migrants. Additional data collection is likely not feasible at this stage; however, the paper could reposition this mechanism as a tentative part of the framework.

We thank R1 for this comment. We agree that better measures of contact extent and intensity would have been highly valuable. We now explicitly reposition the contact mechanism as tentative/exploratory, both in the conceptual framing and in the interpretation of results, as suggested by R1 (see revised framing in the Introduction, lines 67f. and lines 158ff., and the reworked Discussion paragraph, lines 522ff.).

At the same time, we broadened our introduction to contact theory in the context of permanent rural-to-rural environmental migration, as we believe this offers valuable insights to future research. Notably, we study a context where rural-to-rural permanent environmental migration is, since long, ongoing. This seems transferrable to other research, as studies of climate-induced migration (e.g., sea level rise) resemble our case. At the same time, this indicates limits of inference from coarse proxies of intergroup contact theory in contexts with (environmental) in-migration processes ongoing over long time periods. We hence note (see Introduction, end of second to last paragraph): *This proxy offers limited leverage for adjudicating contact theory in a setting where rural-to-rural in-migration is long-standing, because it captures perceived in-migration rather than the frequency and quality of interpersonal interaction, and it does not isolate contact with environmental migrants specifically.*

We extend this in the framework presentation as: *In cases of ongoing environmental changes, we posit [the contact] mechanism as only partly testable, however. This is, as, when rural-to-rural internal (environmental) migration is widespread, researchers observe little to no actual variation regarding the extensive margin of contact (no vs. any contact with (environmental) migrants). This is especially the case where severe (environmental)*

changes are ongoing over longer time spans, as in our case. Then, most localities will have received some extent of (environmental) in-migration. What we can observe, then, are differences in perceptions of actual in-migration, and self-reported intensity of contact (e.g., inclusion of migrants in friendship networks). Both have been shown to shape attitudes in past research (Rudolph and Wagner, 2022; Steinmayr, 2021).

While limited, we then continue to discuss our findings on this, emphasized as tentative in line with R1's request, in the reworked discussion section.

We then end with recommendations for future research. As noted by R1, while we did not consider explicit measures for specific contact with *environmental* as compared to other migrants when planning our research, we see this as worthwhile for future research. Hence, we note this as an intriguing avenue for future research in the discussion section. While most actual permanent in-migration in this area is likely environmental, we again acknowledge that this is a crude measure, however, when wanting to assess how contact shapes the acceptability of migrants with environmental vs. economic motivations. Future research should investigate this question more explicitly, for example by inquiring the extent of respondent's contact with migrants who moved for different motivations. This might be a fruitful extension, given respondents seemingly generalize from the motivations of migrants they come in contact with to group motivations, and consequently to attitudes to these groups in other contexts (Bilgen et al., 2023).

Of course, future research could also explicitly draw on settings where the full extent of real-world variation in exposure with (environmental) migrants is realistic. Usually, this will, however, not be proximate rural areas. These proximate rural areas will usually all be treated once a relevant extent of environmental out-migration occurs. Then, at least some extent of contact will have occurred for all geographic areas, and a thorough test of the contact mechanism becomes impossible.

R1-3 Second, the authors use multiple testing without correction. The manuscript tests 20+ hypotheses across multiple subgroups without any correction for multiple comparisons. This is particularly problematic as several of the key findings rest on marginal p-values that would not survive standard corrections – e.g., H3 (deservingness moderates religious penalty): $p=.146$, H4 experiential proximity (the key empathy finding) is at $p=.122$ (line 354). This is important especially since the title claims 'empathy informs acceptance' but rests on $p=.122$ evidence. The paper need not undertake extensive statistical corrections, but the authors should apply multiple testing corrections and report which findings remain robust and/or consider moderating claims as exploratory than confirmatory.

We thank R1 for this comment and agree that reporting on potential issues with multiple testing will aid readers to understand the robustness of our results and thereby increases the transparency of the validity of our findings.

What R1 rightly highlights is that some of our findings, especially those which we discuss as borderline significant, might actually be false positives. Using this lead, we suggest directly addressing this comment with the concept of the False Discovery Rate (Benjamini and Hochberg, 1995): i.e., what is the probability that a p-value reported as ‘significant’ is actually a false-positive result? To do so, we collect the p-values for all the hypothesis tests reported in Figures 1-4, which we present in the main manuscript. This concerns the p-values for: the seven estimates for the AMCEs for each conjoint level (Figure 1 and numerical in model 1 of SI Table B.4, for H1 and H2); the five estimates for the differences between AMCIEs for the erosion vs. the economic level (Figure 2 and numerical in model 1 of SI Table B.6, for H3); and the four estimates for differences between marginal means for the erosion level between respondent subgroups (Figure 3 and 4 and numerical in Supplementary Tables B.7 and B.8, for H4 and H5). We provide this explanation now also under the newly added Methods Section ‘Corrections for multiple testing’.

These are 16 tests in total – note that this differs from what R1 summarized as the number of tests. This is, as we report in Figure 2 also on the AMCIEs between reason for migration family and economic, to be fully transparent on all our data, even though we do not interpret this substantively (and did not theorize on any differences here). This is also, as we report in Figures 3 and 4 not only on the marginal means for erosion vs. economic, but on the full conjoint table to be transparent – even though we do not theorize on differences beyond erosion vs. economic. Hence, we see these additional reported coefficients rather in the sense of ‘controls’ whose coefficient estimates are shown, but which should not be corrected for in multiple hypothesis testing.

The p-values for these 16 tests are then collected in new SI Table B.8 (see also identical Table RM.1 in this memo). To gauge multiple testing issues with these p-values we follow the Benjamini-Hochberg procedure (Benjamini and Hochberg, 1995) and note which tests retain an interpretation as ‘significant’ under pre-specified false discovery rates (FDR) the reader would have to accept when interpreting the result. Hence, columns 5 to 8 indicate under which tolerable level for the expected proportion of rejections that are type I errors (false rejections) ‘significance’ would have to be interpreted.

In summary, our main results all hold. However, for all results which are only borderline significant – in particular the finding that deservingness moderates religious penalty, that respondents express experiential proximity, and that occupational bias is lessened for environmental migrants only hold if we are willing to accept a high False Discovery Rate of 0.2 (for the latter two) or even 0.3 (for the former). Note, however, that this interpretation is not too different from the already cautious interpretation we gave in the manuscript,

given all these findings are at or above conventional significance thresholds of $p=0.1$ (given our small sample size).

We now provide this interpretation to readers in the manuscript in lines 390-398:

Lastly, in estimating the main AMCEs, the AMCIEs by reason for migration, and the differences between them, as well as marginal means and subgroup differences, we conduct a multitude of statistical tests. We therefore correct for multiple testing using the Benjamini–Hochberg procedure (Benjamini and Hochberg, 1995). The results of these corrections are summarized in Supplementary Table B.8. In short, most of the main interpretations regarding H1 and H2 hold when accepting a False Discovery Rate of 0.05, and a False Discovery Rate of 0.1 regarding H3. However, for our direct evidence regarding H4, experiential proximity, we would have to accept a lenient False Discovery Rate of 0.2 (for details, see extensive interpretation in Methods Section 5.6).

The detailed interpretation is presented in Methods Section 5.6, lines 802-816: *In detail, we interpret Supplementary Table B.8 as follows: Overall, results show that our main deservingness finding is robust to corrections for multiple testing under strict FDRs of 0.05 and 0.1. The finding that environmental migrants receive a weaker penalty for a distant origin compared to economic migrants is robust under an FDR of 0.1. Two additional findings are only supported under a more lenient FDR of 0.2: first, that occupational penalties (teacher vs. cobbler) are attenuated for environmental relative to economic migrants; and second, that prior exposure to erosion (experiential proximity) moderates acceptance of environmental migrants. Finally, the attenuation of religious penalties (lenient Muslim vs. Hindu) for environmental versus economic migrants is supported only under an FDR of 0.3. The fact that these moderating effects hold only under relatively lenient FDR thresholds is consistent with the reported and substantively meaningful differences that, however, mostly fail to reach conventional levels of statistical significance, potentially due to a limited sample size, but potentially also because these are actually false positive results. We therefore interpret these findings with caution, and call on future research to re-test these in fresh samples with appropriate statistical power.*

Last, we agree with R1 that the manuscript’s title, ‘empathy informs acceptance’, rests (in part) on $p=0.122$ significance, which we can only interpret substantively if we are willing to accept a false discovery rate of 0.2. We considered changing or toning down the title, but suggest sticking with it. This is as: first, the (robust) finding that, averaging over occupation, distance, and religion, environmental migrants are more favorably assessed compared to economic migrants, is directly in line with our proposed mechanism, empathy. This is likewise the case for the (robust) finding that various biases are lower for environmental compared to economic migrants. As a third piece to the puzzle, the coefficient for experiential proximity is substantively relevant. Altogether, this suggests to us that empathy seemingly informs acceptance of

environmental migrants for our case, even though for the ‘third piece of the puzzle’ we have to accept an FDR of 0.2. If advised differently, we are happy to change the title, of course.

Again, we thank R1 for recommending this improvement as we think that it allows the reader to more transparently judge the statistical significance of our reported findings.

R1-4 With these remaining clarifications and adjustments, the manuscript would be well-positioned for publication.

Overall, we once again cordially thank R1. The revisions suggested by R1 have helped us to considerably strengthen the contributions of our manuscript.

Estimate	Attribute	Attribute level	p-value	FDR 0.05	FDR 0.1	FDR 0.2	FDR 0.3
AMCE	reason	family	0.000	✓	✓	✓	✓
AMCE	reason	erosion	0.000	✓	✓	✓	✓
AMCE	occupation	teacher	0.000	✓	✓	✓	✓
AMCE	occupation	farmer	0.004	✓	✓	✓	✓
AMCE	religion	strict muslim	0.000	✓	✓	✓	✓
AMCE	religion	lenient muslim	0.000	✓	✓	✓	✓
AMCE	origin	distant	0.066			✓	✓
AMCIE Difference (erosion - economic)	occupation	teacher	0.095			✓	✓
AMCIE Difference (erosion - economic)	occupation	farmer	0.294				
AMCIE Difference (erosion - economic)	religion	strict muslim	0.739				
AMCIE Difference (erosion - economic)	religion	lenient muslim	0.143				✓
AMCIE Difference (erosion - economic)	origin	distant	0.025		✓	✓	✓
MM Difference (erosion perceived as main event: Yes - No)	reason	erosion	0.802				
MM Difference (new families arrived: Yes - No)	reason	erosion	0.864				
MM Difference (previously migrated: Yes - No)	reason	erosion	0.795				
MM Difference (lost house due to erosion: Yes - No)	reason	erosion	0.122			✓	✓

Table RM.1. *P-values for main hypothesis tests that are reported in the manuscript and whether p-values hold against thresholds accounting for different false-discovery rates (FDR) according to the Benjamini-Hochberg procedure. We sort p-values in ascending order, specify the total number of tests, and compare each p-value to a corresponding threshold that depends on its rank in the ordered list, the number of tests, and the specified FDR, where the FDR is the tolerated expected proportion of rejections that are type I errors (false rejections). A p-value is considered to retain significance under a given FDR level if it, or any p-value with a smaller rank (i.e., appearing earlier in the ordered list), is lower than its corresponding threshold for that FDR level.*

Comments by Reviewer #2

R2-1 I am very satisfied with the way the authors have addressed all the suggested points. It is evident that they have invested considerable effort in developing this research, and the outcome is commendable. The study's contribution is clear, and I am confident that both academic scholarship and the policy sector will benefit significantly from its findings.

Overall, we once again thank R2 very much. The revisions suggested by R2 have substantially strengthened the contributions of our manuscript.